# Gaussian-Augmented Physics Simulation and System Identification with Complex Colliders

**Federico Vasile**[1]    **Ri-Zhao Qiu**[2]    **Lorenzo Natale**[1]    **Xiaolong Wang**[2]
[1]Istituto Italiano di Tecnologia, [2]UC San Diego
https://as-diffmpm.github.io

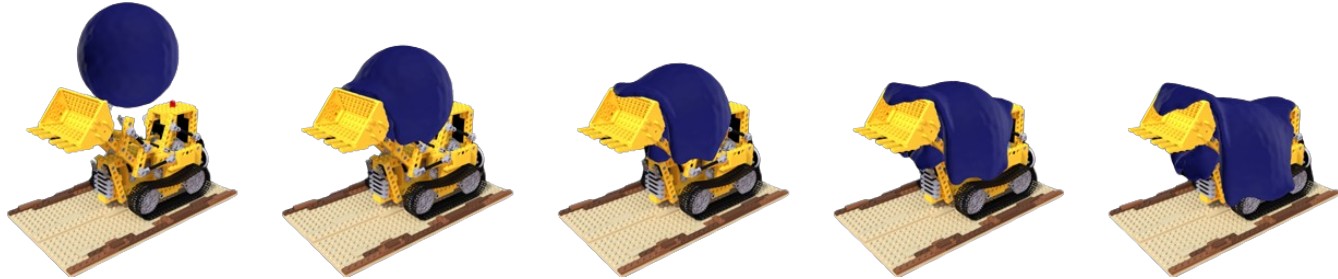

Figure 1: We present the **Any-Shape Differentiable Material Point Method** (**AS-DiffMPM**), a particle-based framework for simulating collisions with **arbitrarily shaped** rigid bodies. When integrated with Gaussian-based rendering, it enables physically plausible **animation** and **system identification** (estimation of object physical parameters from visual observations).

## Abstract

System identification involving the geometry, appearance, and physical properties from video observations is a challenging task with applications in robotics and graphics. Recent approaches have relied on fully differentiable Material Point Method (MPM) and rendering for simultaneous optimization of these properties. However, they are limited to simplified object-environment interactions with planar colliders and fail in more challenging scenarios where objects collide with non-planar surfaces. We propose AS-DiffMPM, a differentiable MPM framework that enables physical property estimation with arbitrarily shaped colliders. Our approach extends existing methods by incorporating a differentiable collision handling mechanism, allowing the target object to interact with complex rigid bodies while maintaining end-to-end optimization. We show AS-DiffMPM can be easily interfaced with various novel view synthesis methods as a framework for system identification from visual observations.

## 1 Introduction

*Bending but not breaking*, a branch dances with the wind—yielding to its force yet never surrendering. From the gentle drift of a balloon to the violent shatter of glass, the world reveals its hidden truths through motion. We, too, learn not only by *seeing* but by *understanding*—tracing back from movement to the forces that shaped it. In this work, we give machines this same power: *to see, reason, and uncover the physical essence* of objects through their dance with the world, recovering their hidden properties from mere pixels.

Estimating the physical parameters of objects from visual observations is a challenging task in computer vision, with applications in robotics, virtual reality, and graphics. While existing methods

39th Conference on Neural Information Processing Systems (NeurIPS 2025).

such as PhysGaussian [1] can **animate** static scenes by simulating plausible dynamics, they do not support the **identification** of object properties from dynamic scenes. Humans naturally infer an object's geometry and dynamics through a two-step process. First, they recognize and understand its shape (e.g., a ball). Then, by observing its interactions with the environment (e.g., let the ball fall on the ground), they deduce its physical properties. Inspired by this intuition, modern system identification methods follow a similar paradigm, combining two components: a *rendering model* to reconstruct geometry from images and a *physics engine* to simulate motion, enabling the estimation of physical properties from video sequences.

Recent advances in scene representations — particularly point-based models [2]—and particle-based physics simulation [3], have facilitated the animation of static geometry with physics-grounded motion. This process involves establishing correspondences between the points used for rendering and the particles in simulation, allowing the radiance field to be animated by applying the motion derived from simulated particle trajectories. As a result, for physical properties estimation, this approach can be applied by reconstructing the object's geometry, initializing an estimate of its physical parameters, and refining them through gradient-based optimization, comparing simulated motion against real-world observations.

PAC-NeRF [4] is a pioneering model for system identification from visual observations; combining voxel-based radiance fields [5] with a Differentiable Material Point Method (DiffMPM) [6] to enable gradient-based optimization of physical parameters from images. However, this approach, along with other recent works [7, 8, 9], share a significant limitation: they perform system identification only when the object interacts with simple boundaries such as the ground. While effective, this setup fails to capture more complex scenarios, where objects interact with arbitrarily shaped rigid bodies.

In this work, we extend system identification beyond simple boundary interactions, enabling estimation of physical properties in more complex settings. To the best of our knowledge, this problem has not been previously addressed, primarily due to two key limitations: (i) existing system identification methods are restricted to interactions with a planar boundary [4, 7, 8, 9], and (ii) available MPM-based simulators that accurately handle arbitrarily shaped colliders are not differentiable [10], thus not suitable for physical parameters optimization. Our contribution is threefold:

1. **Any-Shape Differentiable MPM.** We propose AS-DiffMPM, a Differentiable MPM capable of handling collisions between the target object and arbitrarily shaped colliders. Quantitative experiments on *Newtonian*, *non-Newtonian* and *granular* materials demonstrate the necessity of an ad hoc solution for rigid body interactions, which previous system identification methods have overlooked.

2. **Versatile Collision Handling Framework.** We design a general interface to integrate colliders of various formats (i.e., mesh and 2D Gaussians [11]) into AS-DiffMPM.

3. **Rendering-Physics Integration for System Identification.** We combine our AS-DiffMPM with multiple rendering models and establish a benchmark for system identification from visual observations when the object collides with arbitrarily shaped rigid bodies.

## 2 Related Work

**Scene Representations for Photorealistic Rendering.** The topic of scene reconstruction and synthesizing photorealistic novel views has been studied in the computer vision community for decades [12, 13, 14]. Recently, there has been a growing interest in using learning-based differentiable rendering or rasterization for such a purpose [2, 15, 16, 17, 18]. These methods are often categorized into implicit rendering methods based on ray-marching queries [15, 16, 17, 18] or explicit rasterization methods [2, 11]. To manipulate and animate these static scene representations, researchers have shown that it is possible to edit both implicit methods (often via remapping voxel-based representations [4, 19]) and explicit methods (via direct manipulation of explicit units [8, 1, 20]). Our method is mostly orthogonal to the development in scene representations. The AS-DiffMPM is a plug-and-play module to both NeRF- and 3DGS-based methods, as quantitatively shown in the experiment.

**Physics-integrated Scene Synthesis.** While high-quality reconstruction methods offer a pathway to synthesize *static* scenes from a set of images in novel viewpoints, they are limited to understanding static appearances. Humans, on the other hand, have the ability to picture physically plausible interactions with the scenes. To equip machines with such an intelligence, researchers have sought to animate static scenes with physics simulators to generate animated scenes that are both visually and

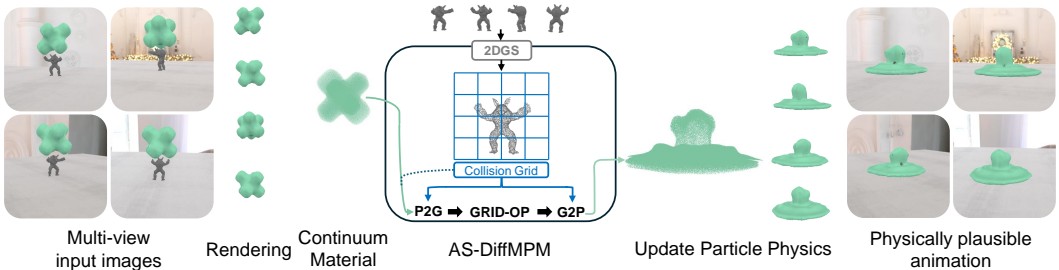

Multi-view input images    Rendering    Continuum Material    AS-DiffMPM    Update Particle Physics    Physically plausible animation

Figure 2: **Overview.** Given multi-view input images, we separately reconstruct the continuum object using a rendering model (e.g., [5, 11]) and the rigid body collider with 2DGS [11]. Particle trajectories are subsequently advected using AS-DiffMPM. Finally, the updated particle positions are mapped back to rendering primitives for photorealistic and physically plausible animation.

physically appealing [1, 21, 20, 22, 4, 23, 24]. Due to the indirection caused by implicitness, few methods animate implicit NeRFs [23]. Many researchers have combined Material Point Method [3] with Gaussian Splatting [2] due to their naturally aligned point-based representations. These methods have shown to provide photorealistic animations [1, 25] with extensions to VR applications [22], language-driven animations [20], and complex fluid dynamics [21]. However, these methods provide physically plausible *animations* **of static scenes** rather than estimating physical parameters of objects. Another line of work leverages object dynamics priors from video generative models to endow **static 3D objects with *interactive* behaviors** [26, 27, 28, 29]. While effective for visually realistic interactions—useful in applications such as content creation—they may fail in domains like robotics, where precise physical understanding is necessary. In contrast, our work aims to recover accurate physical parameters of objects from visual observations.

**System Identification from Videos.** The identification of dynamic systems goes beyond simple animation, as it requires not only visually plausible animations but also accurate physical properties that replicate the behavior of the reference system. Prior to the introduction of photorealistic differentiable rendering [15], system identifications from videos have been difficult due to the coupling of appearance and dynamics [30]. Recent works have leveraged gradient-based optimization of physical parameters through differentiable rendering and simulation for system identification. PAC-NeRF [4] is a representative method that combines a voxel-based NeRF [5] and DiffMPM for this task. Following up to this work, GIC [8] introduces a novel scene representation method upon 3DGS [2] to improve the supervisory signal from visual observations. Additionally, [9] replaces MPM with a Spring-Mass representation to model elastic objects. However, these methods focus on system identification under the assumption of object interactions with planar surfaces, as the underlying physics simulator does not support complex colliders. Our AS-DiffMPM bridges this gap by supporting system identification under more realistic conditions, such as collisions with complex colliders.

## 3 Preliminary

### 3.1 Material Point Method

The Material Point Method (MPM) [31, 32, 3] is a widely adopted approach to simulate the dynamics of continuum materials. It combines the strengths of both Lagrangian particles and Eulerian grid by discretizing the continuum as a set of particles carrying physical information such as position, velocity, and mass. For every update step, these properties are transferred to the Eulerian background grid to solve the equations of motion on the grid nodes. Finally, the resulting nodal velocities on the grid nodes are interpolated back to the Lagrangian particles, thus updating the state of the material. MPM operates in three stages: *Particle-to-Grid (P2G)* transfer, *Grid Operations (G-OP)*, and *Grid-to-Particle (G2P)* transfer.

Specifically, considering a toy example where an object is subject to gravity, we represent the object as a set of particles carrying the following Lagrangian quantities: the particle position $\mathbf{x}_p$, velocity $\mathbf{v}_p$, and mass $m_p$. These particles interact with an Eulerian grid, which stores nodal quantities: the position $\mathbf{x}_g$, mass $m_g$ and velocity $\mathbf{v}_g$ at the grid node $g$.

At each time step, the simulation proceeds through the following stages:

- *Particle-to-Grid (P2G)*: Physical quantities from particles are transferred to nearby grid nodes using a weighting function $w_g(\mathbf{x}_g - \mathbf{x}_p)$. The mass and momentum of each particle contribute to the grid:

$$m_g = \sum_p m_p w_g(\mathbf{x}_g - \mathbf{x}_p), \quad \mathbf{v}_g = \frac{1}{m_g} \sum_p m_p \mathbf{v}_p w_g(\mathbf{x}_g - \mathbf{x}_p). \tag{1}$$

- *Grid Operations (G-OP)*: Once the mass and momentum fields are constructed on the grid, forces such as gravity $\mathbf{g}$ are applied to update velocities at the grid nodes: $\mathbf{v}_g \leftarrow \mathbf{v}_g + \Delta t \mathbf{g}$. Next, boundary conditions are enforced to ensure that the object correctly interacts with obstacles in the scene. In this example, we consider a *sticky ground*, meaning any particle colliding with the surface loses velocity upon impact. The ground is represented as a plane defined by a point $\mathbf{x}_b$ and a normal $\mathbf{n}_b$. A grid node $g$ is in contact with the ground if $(\mathbf{x}_g - \mathbf{x}_b) \cdot \mathbf{n}_b \leq 0$, in which case its velocity is set to zero, i.e., $\mathbf{v}_g = \mathbf{0}$.
- *Grid-to-Particle (G2P)*: The velocity field from the grid nodes is interpolated back to the particles to update their state. Each particle gathers velocity from the surrounding grid nodes:

$$\mathbf{v}_p = \sum_g \mathbf{v}_g w_g(\mathbf{x}_g - \mathbf{x}_p). \tag{2}$$

The particle positions are then updated: $\mathbf{x}_p \leftarrow \mathbf{x}_p + \Delta t \mathbf{v}_p$.

### 3.2   System identification

System identification aims to estimate the geometric structure and physical properties of dynamic objects from multi-view video sequences. Following prior works [4, 7, 8], we assume that the object material (e.g., Newtonian) is known and that its behavior adheres to continuum mechanics [32, 33].

## 4   Rigid Body Colliders

### 4.1   Collision Resolution in MPM

A common approach for handling collisions between the continuum material and a rigid body is to apply velocity corrections during the *G-OP* stage, as introduced in Sec. 3.1. Specifically, level sets (or signed distance functions) are used to classify each grid node as inside or outside the collider and adjust the nodal velocity accordingly [31]. Although straightforward and effective, this method fails with complex geometries (e.g., open surfaces, sharp boundaries), since all the particles in the same grid-cell region share a single velocity field. In contrast, Compatible Particle-in-Cell (CPIC) [10] handles collisions in a particle-wise manner during the *P2G* and *G2P* steps. It partitions grid nodes and particles into compatible or incompatible sets relative to the boundary's surface, allowing more precise velocity corrections for each particle colliding with the rigid body.

Hereafter, we detail the CPIC [10] method to project the rigid body onto a grid-based distance field (Sec. 4.1.1). This information is transferred to the continuum material (Sec. 4.1.2) and used to possibly apply velocity corrections on particles based on their relationship with neighboring grid nodes (Sec. 4.1.3).

#### 4.1.1   Project Rigid Body to Collision Grid

Following prior work [10], we represent the rigid body collider as a mesh and project it onto a grid that encodes properties for collision handling. We define a Collision Grid with the same resolution as the MPM Eulerian grid and store the following quantities at each grid node: (i) the *affinity* flag $A_g$ indicating whether the grid node $g$ is near the boundary, (ii) the *unsigned distance* $d_g$ from the grid node to the boundary, (iii) a *tag* $T_g$ denoting which side of the boundary the grid node is on and (iv) a *normal* $\mathbf{n}_g$ retrieved from the boundary. The projection process follows three key steps: sampling rigid particles, identifying affinity nodes and reconstructing the Collision Grid.

**Sampling rigid particles.** To map the rigid body's surface onto nearby grid nodes, the collider is treated as a collection of primitives, thus, we sample a predefined number of rigid particles on each

face $\xi^i$ of the mesh representing the rigid body. Moreover, we define $\xi(\mathbf{x}_{rp})$ as the face to which the rigid particle $\mathbf{x}_{rp}$ belongs.

**Identifying affinity nodes.** In order to identify all the grid nodes surrounding the rigid body, we map each $\mathbf{x}_{rp}$ to the $3\times3\times3$ grid of neighboring grid nodes denoted as $\mathcal{N}(\mathbf{x}_{rp})$. For each grid node $g \in \mathcal{N}(\mathbf{x}_{rp})$, its position $\mathbf{x}_g$ is projected onto the plane defined by the face $\xi(\mathbf{x}_{rp})$. If the projection falls inside the face, we store the $(\mathbf{x}_g, \xi(\mathbf{x}_{rp}))$ pair and set $A_g = 1$ (otherwise, we set $A_g = 0$).

**Reconstructing the Collision Grid.** Since there may be multiple faces in proximity to the same affinity grid node, i.e., $\{(\mathbf{x}_g, \xi^i)\}, i \in \{1, \ldots, l\}$, we select the one with minimum unsigned point-plane distance:

$$\xi_g^* = \underset{i \in \{1,\ldots,l\}}{\arg\min} |dist(\mathbf{x}_g, \xi^i)|, \tag{3}$$

and store its normal in $\mathbf{n}_g$, the sign of the distance in $T_g = sign(dist(\mathbf{x}_g, \xi_g^*))$ and the unsigned distance in $d_g$.

In summary, for each grid node $g$, the Collision Grid carries the following properties: $A_g, d_g, T_g, \mathbf{n}_g$.

### 4.1.2 Transfer Collision Grid to Material Particles

The Collision Grid is used to transfer properties to material particles, ensuring accurate interactions with the boundary and possibly correcting undesired behaviors (e.g., penetration).

**Particle affinity.** Given the material particle position $\mathbf{x}_p$, we set $A_p = 1$ if at least one of the $3\times3\times3$ neighboring grid nodes $g \in \mathcal{N}(\mathbf{x}_p)$ has $A_g = 1$.

**Particle distance, tag, and normal.** For each material particle $\mathbf{x}_p$ in affinity with the Collision Grid, we retrieve the collision properties through interpolation over $\mathcal{N}(\mathbf{x}_p)$:

$$d_p = \sum_{g \in \mathcal{N}(\mathbf{x}_p)} w_g(\mathbf{x}_g - \mathbf{x}_p) A_g T_g d_g, \quad \mathbf{n}_p = \sum_{g \in \mathcal{N}(\mathbf{x}_p)} w_g(\mathbf{x}_g - \mathbf{x}_p) A_g \mathbf{n}_g, \tag{4}$$

where $w$ is a nodal weighting function (e.g., B-spline [31, 34]). Then, we set $T_p = sign(d_p)$. However, if a particle penetrates the boundary, an incorrect $T_p$ may be reconstructed. Therefore, each particle's tag $T_p$ retains its first acquired value until the particle loses affinity, i.e., moves away from the boundary. As a result, if a particle penetrates the boundary due to simulation errors, it will have an incorrect $d_p$ (i.e., wrong sign) while still retaining a correct $T_p$. This allows penetrations to be fixed: $\mathbf{f}_p = -k_h d_p \mathbf{n}_p$, where $\mathbf{f}_p$ is the penalty force to apply to the material particle $p$ and $k_h$ is the penalty stiffness parameter.

**Particle-grid compatibility.** When a material particle approaches the boundary, its tag $T_p$ is compared with the neighboring nodes $g \in \mathcal{N}(\mathbf{x}_p)$: $p$ and $g$ are incompatible if $T_p = 1$ and $T_g = -1$, or vice versa. In other words, they are on two different sides of the boundary. This condition is used during the *P2G* and *G2P* (Sec. 4.1.3) to identify the particles that are about to collide and adjust their velocities in a particle-wise manner, rather than applying nodal velocities during the *G-OP* stage.

### 4.1.3 P2G and G2P with CPIC

During the *P2G* stage, material particles transfer velocities only to compatible grid nodes. Consequently, no collision handling occurs during the *G-OP* stage, as nodal velocities receive contributions solely from particles away from the boundary. In the *G2P* stage, however, material particles gather velocities from both compatible and incompatible grid nodes. For each incompatible node, we directly reuse the particle's current velocity $\mathbf{v}_p$, enabling particle-wise collision handling with the surface. For instance, assuming a slippery surface, the velocity is projected onto the surface as $\mathbf{v}_p^{proj} = \mathbf{v}_p - (\mathbf{v}_p \cdot \mathbf{n}_p)\mathbf{n}_p$.

### 4.2 Representing Rigid Body as 2D Gaussians

As discussed in Sec. 4.1.1, previous work represents the rigid body collider as a mesh [10]. In contrast, we design a generalized interface for our framework, enabling seamless integration of both meshes and 2D Gaussians [11] as colliders. By utilizing the Collision Grid as a proxy for transferring collision properties from the rigid body to the material particles, our framework can accommodate any collider represented as a set of primitives $\xi^i$ with associated normals $\mathbf{n}_i$.

We reconstruct the rigid body from multi-view images using 2D Gaussian Splatting [11]. Thus, in this case, the rigid body primitive $\xi^i$ corresponds to a planar disk (i.e., 2D Gaussian) rather than a mesh face. The collider is imported into AS-DiffMPM following the procedure in Sec. 4.1, with minimal adaptation required for identifying the affinity grid nodes (Sec. 4.1.1).

**Sampling rigid particles**. We follow the previously described procedure to sample rigid particles $\mathbf{x}_{rp}$ on the primitives $\xi^i$.

**Identifying affinity nodes.** The $3{\times}3{\times}3$ neighboring grid nodes $\mathcal{N}(\mathbf{x}_{rp})$ of rigid particle $\mathbf{x}_{rp}$ are projected onto the plane defined by the planar disk $\xi^i$. If the projection falls inside the disk, we store the $\big(\mathbf{x}_g, \xi(\mathbf{x}_{rp})\big)$ and set the affinity flag $A_g$ accordingly.

From this stage onward, the procedure adheres to the presented approach for transferring collision properties to material particles and resolving collisions during the *P2G* and *G2P* stages.

## 4.3 System Identification

We integrate our AS-DiffMPM with multiple rendering methods to enable system identification from visual observations.

**Voxel-based NeRF.** We follow [4] to integrate a voxel-based NeRF [5] with AS-DiffMPM. Both MPM and voxel-based NeRFs rely on a grid view $(G)$ for computation and rendering, respectively. However, while MPM uses a Lagrangian view $(P)$ to advect particles representing the continuum material, voxel-based NeRFs lack an equivalent point-based representation. Thus, a mapping between voxels and Lagrangian particles is required to enable dynamic rendering. This process involves: (i) mapping voxel fields $\mathcal{F}_i^G$ to Lagrangian particle fields $\mathcal{F}_p^P$, (ii) advecting particles using MPM and (iii) mapping the updated particle positions back to voxels for dynamic rendering. The following interconverters for $G$ and $P$ views are used:

$$\mathcal{F}_p^P \approx \sum_i w_{ip}\mathcal{F}_i^G, \ \mathcal{F}_i^G \approx \frac{\sum_p w_{ip}\mathcal{F}_p^P}{\sum_p w_{ip}}, \tag{5}$$

where $p$ and $i$ indicate the index of the particle and grid node, respectively; and $w_{ip}$ is a weighting function defined on $i$ and evaluated at $p$. As a result, this mapping mechanism enable bidirectional transfer between the voxel grid and Lagrangian particles, allowing for gradient-based optimization of physical parameters from visual observations.

**Point-based methods.** Point-based rendering methods [8, 2, 11] naturally integrate with MPM as they both utilize the Lagrangian view $(P)$ for rendering and particle advection. By employing differentiable point-based primitives [2, 11], the gradient of the photometric loss with respect to each rendering primitive is computed and propagated to its corresponding Lagrangian particle, thereby supporting gradient-based optimization of the physical parameters from visual observations. Formally:

$$\frac{\partial\mathcal{L}}{\partial\theta} = \sum_r \sum_p \underbrace{\left(\frac{\partial\mathcal{L}}{\partial I} \cdot \frac{\partial I}{\partial\mathbf{x}_r}\right)}_{\text{Rendering}} \cdot \underbrace{\frac{\partial\mathbf{x}_r}{\partial\mathbf{x}_p^r}}_{\text{Mapping}} \cdot \underbrace{\frac{\partial\mathbf{x}_p^r}{\partial\theta}}_{\text{MPM}}, \tag{6}$$

where $\mathcal{L}$ is the photometric loss comparing the rendered image $I$ with the ground truth, $\mathbf{x}_r$ denotes the location of the rendering primitive $r$, $\mathbf{x}_p^r$ indicates the position of the Lagrangian particle $p$ associated with primitive $r$ and $\theta$ represents the physical parameters (e.g., stiffness, viscosity). The final gradient is the product of three components: (i) rendering, (ii) primitive-particle mapping, and (iii) differentiable MPM. For simplicity, each Lagrangian particle is initialized from—and tied to—a rendering primitive (i.e., $\mathbf{x}_p^r = \mathbf{x}_r$), reducing the gradient flow to:

$$\frac{\partial\mathcal{L}}{\partial\theta} = \sum_p \left(\frac{\partial\mathcal{L}}{\partial I} \cdot \frac{\partial I}{\partial\mathbf{x}_p}\right) \cdot \frac{\partial\mathbf{x}_p}{\partial\theta}. \tag{7}$$

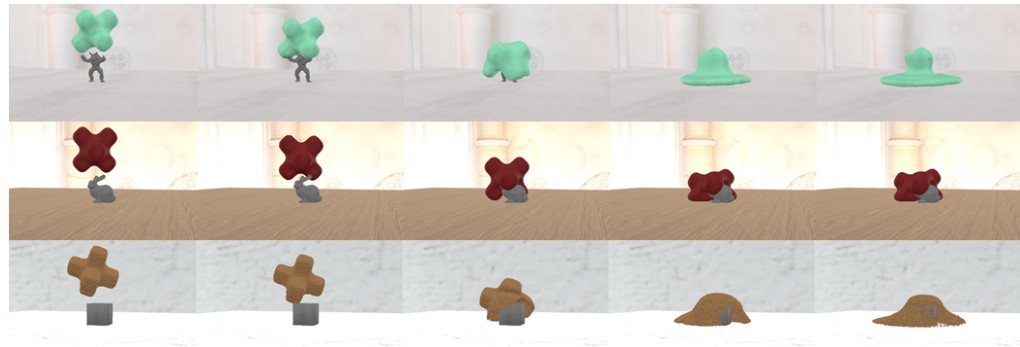

Figure 3: Qualitative examples of reference frames used in experiments. Note that previous works [4, 7, 8, 9] perform system identification only with a planar surface and do not support such colliders.

## 5 Experiments

**Physical Parameters.** Our analysis focuses on material types that undergo noticeable deformation upon collision with rigid bodies, specifically Newtonian fluids, non-Newtonian fluids, and granular media. For Newtonian fluids, we estimate fluid viscosity ($\mu$) and bulk modulus ($\kappa$). For non-Newtonian fluids, we recover shear modulus ($\mu$), bulk modulus ($\kappa$), yield stress ($\tau_Y$), and plasticity viscosity ($\eta$). For granular media, we estimate the friction angle ($\theta_{fric}$). For a comprehensive overview of physical parameters and constitutive models, we refer to [4].

**Dataset.** We follow the protocol in [4] and generate ground-truth simulation rollouts using the cross-shaped object from [4] as our continuum material, with the physical parameter values provided in Appendix E. The object undergoes free fall and collides with a static rigid body (*Box*, *Bunny*, or *Armadillo*) having sticky surfaces. For each rollout, we collect both the 3D particle trajectories (i.e., point clouds over time) of the continuum material and the corresponding rendered frames (see Fig. 3), captured from 11 cameras placed around the scene. These data are used in Sec. 5.1 and Sec. 5.2 for experiments on system identification. Following the protocol in [4], we generate 10 rollouts per collider for each material type, except for granular media, where 5 videos are generated, resulting in a total of 75 rollouts. Each rollout is 16 timesteps long.

**Training and Evaluation.** Our training setup also adheres to [4], including the use of the Adam optimizer [35] and the initial guesses for the physical parameters. Unlike [4], we do not optimize the initial velocity vector during the free-fall phase; instead, we use the ground truth values, as our focus is on system identification during collisions. Finally, we report the results of physical parameters estimation using the mean and standard deviation of absolute errors, scaled by a factor of 100.

### 5.1 System Identification from Particle Trajectories

**Training.** This experiment evaluates system identification performance by estimating the physical parameters of the continuum material using the 3D particle trajectory as supervision. Specifically, we compute a particle-wise Mean Squared Error (MSE) loss between the reference trajectory and the one simulated by the MPM using the current parameter estimates. The loss is backpropagated through the differentiable simulator to iteratively optimize the physical parameters. We emphasize that this experiment does not involve visual observations or rendering models, as the focus is exclusively on evaluating the effectiveness of different collision handling strategies for system identification.

**Baselines.** As discussed in Sec. 4.1, a straightforward approach for collision handling in MPM is during the *G-OP* stage, where a Signed Distance Function (SDF) is reconstructed from the rigid body collider [27]. Assuming sticky surfaces, the nodal velocities for grid nodes inside the collider are set to zero. We refer to this baseline as GOP-DiffMPM. Additionally, following prior works [1, 36], the rigid collider can be represented as a set of rigid particles. We denote this approach as RP-DiffMPM.

**Results.** Tab. 1 compares the performance of AS-DiffMPM with baselines. Notably, no consistent trend is observed across colliders: even geometrically simpler shapes like the Box can present

| Material | Collider | Parameters | RP-DiffMPM | GOP-DiffMPM | AS-DiffMPM (**Ours**) |
|---|---|---|---|---|---|
| Newtonian | Box | $\log_{10}(\mu)$ $\log_{10}(\kappa)$ | $4.82 \pm 5.15$ $37.97 \pm 21.8$ | $2.73 \pm 3.95$ $17.09 \pm 19.9$ | $5.08 \pm 2.73$ $27.81 \pm 33.49$ |
| | Bunny | $\log_{10}(\mu)$ $\log_{10}(\kappa)$ | $6.69 \pm 5.97$ $35.04 \pm 37.35$ | $0.43 \pm 0.3$ $11.52 \pm 15.97$ | $4.35 \pm 5.24$ $31.37 \pm 50.02$ |
| | Armadillo | $\log_{10}(\mu)$ $\log_{10}(\kappa)$ | $5.67 \pm 3.98$ $34.61 \pm 46.52$ | $0.83 \pm 1.14$ $13.08 \pm 10.66$ | $0.4 \pm 0.76$ $0.22 \pm 0.24$ |
| Non-Newtonian | Box | $\log_{10}(\mu)$ $\log_{10}(\kappa)$ $\log_{10}(\tau_Y)$ $\log_{10}(\eta)$ | $42.01 \pm 32.76$ $99.55 \pm 98.64$ $20.13 \pm 25.83$ $52.38 \pm 27.63$ | $41.11 \pm 27.78$ $136.04 \pm 78.32$ $5.86 \pm 3.21$ $53.22 \pm 42.22$ | $23.51 \pm 30.07$ $27.06 \pm 19.12$ $7.28 \pm 3.75$ $17.7 \pm 15.17$ |
| | Bunny | $\log_{10}(\mu)$ $\log_{10}(\kappa)$ $\log_{10}(\tau_Y)$ $\log_{10}(\eta)$ | $53.02 \pm 47.05$ $82.83 \pm 86.55$ $31.86 \pm 50.01$ $56.01 \pm 31.13$ | $20.29 \pm 13.96$ $134.82 \pm 75.71$ $7.10 \pm 2.72$ $53.36 \pm 36.15$ | $21.92 \pm 36.89$ $26.77 \pm 25.09$ $2.48 \pm 1.87$ $39.1 \pm 36.2$ |
| | Armadillo | $\log_{10}(\mu)$ $\log_{10}(\kappa)$ $\log_{10}(\tau_Y)$ $\log_{10}(\eta)$ | $36.67 \pm 20.92$ $192.43 \pm 76.69$ $13.72 \pm 4.58$ $64.40 \pm 40.39$ | $36.81 \pm 34.27$ $176.31 \pm 73.89$ $12.10 \pm 5.29$ $54.47 \pm 38.32$ | $7.14 \pm 6.64$ $56.26 \pm 76.97$ $5.07 \pm 5.79$ $44.98 \pm 41.34$ |
| Granular | Box | $\theta_{fric}$ | $1.58 \pm 0.65$ | $1.16 \pm 0.84$ | $1.22 \pm 0.71$ |
| | Bunny | $\theta_{fric}$ | $1.91 \pm 0.42$ | $0.11 \pm 0.15$ | $0.06 \pm 0.07$ |
| | Armadillo | $\theta_{fric}$ | $2.28 \pm 0.66$ | $0.26 \pm 0.16$ | $0.51 \pm 0.96$ |

Table 1: **System identification from particle trajectories.** We compare AS-DiffMPM with two differentiable baselines for collision handling with arbitrarily shaped rigid bodies. Each method is evaluated based on its ability to recover physical parameters using particle trajectories as supervision. Best (light red) and second best (very light red) results are highlighted.

challenges due to sharp corners, while more complex shapes like the Bunny and Armadillo have intricate surfaces leading to complex particle trajectories. Overall, AS-DiffMPM achieves comparable or superior performance, due to its particle-wise collision handling. In contrast, GOP-DiffMPM may be less accurate with complex geometries, as it assigns a single velocity field to all particles within the same grid cell. Finally, RP-DiffMPM consistently underperforms, suggesting that accurate system identification requires tailored collision resolution strategies.

## 5.2 System Identification from Visual Observations

**Training.** We follow the system identification pipeline outlined in [4]. Given a multi-view video, we first segment the continuum object using video matting techniques [37]. Moreover, since our setting involves interactions with a rigid body, we import the mesh-based collider into AS-DiffMPM following the procedure described in Sec. 4.2. The *static* geometry of the continuum object is then reconstructed from the multi-view images at the initial timestep. With the scene setup complete, we initialize the physical parameters with an initial guess and start the optimization process.

**Rendering Models.** To the best of our knowledge, existing system identification methods [4, 7, 8, 9] do not account for scenarios involving continuum materials interacting with arbitrarily shaped rigid bodies. To address this limitation, we integrate AS-DiffMPM with several rendering models and establish a benchmark for this novel setting. Specifically, we combine AS-DiffMPM with the voxel-based NeRF from [5] (DVGO), the point-based 2D Gaussian Splatting method from [11] (2DGS), and the Motion-factorized Dynamic 3D Gaussian Splatting model (MDyn-3DGS) introduced in [8]. Unlike the first two methods, MDyn-3DGS performs dynamic scene reconstruction prior to optimization. This reconstructed information is then used during physical parameter optimization to constrain particles within the rendered dynamic scene.

**Results.** The results presented in Tab. 2 are summarized as follows:

- *Newtonian fluid*: MDyn-3DGS [8] generally provides more accurate estimates of physical parameters compared to 2DGS [11] and DVGO [5]. It also matches or exceeds the performance

| Material | Collider | Parameters | DVGO [5] | 2DGS [11] | MDyn-3DGS [8] |
|---|---|---|---|---|---|
| Newtonian | Box | $\log_{10}(\mu)$
$\log_{10}(\kappa)$ | $6.71 \pm 4.4$
$120.94 \pm 73.37$ | $6.68 \pm 4.55$
$112.13 \pm 85.89$ | $5.58 \pm 3.25$
$40.07 \pm 39.47$ |
| | Bunny | $\log_{10}(\mu)$
$\log_{10}(\kappa)$ | $3.55 \pm 4.36$
$53.34 \pm 64.94$ | $7.37 \pm 5.91$
$99.16 \pm 68.65$ | $3.29 \pm 2.08$
$22.38 \pm 17.32$ |
| | Armadillo | $\log_{10}(\mu)$
$\log_{10}(\kappa)$ | $1.99 \pm 1.59$
$32.94 \pm 42.24$ | $5.28 \pm 2.55$
$49.66 \pm 35.79$ | $0.97 \pm 0.86$
$14.99 \pm 17.13$ |
| Non-Newtonian | Box | $\log_{10}(\mu)$
$\log_{10}(\kappa)$
$\log_{10}(\tau_Y)$
$\log_{10}(\eta)$ | $23.83 \pm 14.77$
$52.76 \pm 51.68$
$5.84 \pm 4.34$
$26.10 \pm 19.21$ | $28.88 \pm 50.41$
$70.78 \pm 88.13$
$12.77 \pm 16.22$
$58.02 \pm 41.46$ | $67.55 \pm 128.49$
$32.10 \pm 49.97$
$8.8 \pm 2.86$
$51.25 \pm 28.81$ |
| | Bunny | $\log_{10}(\mu)$
$\log_{10}(\kappa)$
$\log_{10}(\tau_Y)$
$\log_{10}(\eta)$ | $55.38 \pm 31.85$
$68.84 \pm 38.11$
$11.43 \pm 6.25$
$62.06 \pm 49.71$ | $10.05 \pm 7.95$
$44.44 \pm 33.39$
$3.30 \pm 2.15$
$46.05 \pm 27.02$ | $15.91 \pm 25.49$
$23.16 \pm 18.29$
$4.97 \pm 5.7$
$43.37 \pm 22.43$ |
| | Armadillo | $\log_{10}(\mu)$
$\log_{10}(\kappa)$
$\log_{10}(\tau_Y)$
$\log_{10}(\eta)$ | $79.23 \pm 82.24$
$107.90 \pm 95.32$
$63.01 \pm 148.55$
$83.60 \pm 50.2$ | $12.34 \pm 13.32$
$33.30 \pm 20.60$
$2.34 \pm 2.41$
$55.18 \pm 37.20$ | $12.62 \pm 10.30$
$19.75 \pm 13.67$
$4.32 \pm 1.82$
$39.74 \pm 23.07$ |
| Granular | Box | $\theta_{fric}$ | $4.16 \pm 0.76$ | $0.62 \pm 0.75$ | $3.16 \pm 1.04$ |
| | Bunny | $\theta_{fric}$ | $4.02 \pm 0.68$ | $0.50 \pm 0.29$ | $3.33 \pm 0.62$ |
| | Armadillo | $\theta_{fric}$ | $4.11 \pm 1.05$ | $0.36 \pm 0.13$ | $2.89 \pm 0.78$ |

Table 2: **System identification from visual observations.** We evaluate our AS-DiffMPM combined with three rendering methods on the task of physical parameter estimation from multi-view videos. Best (light red) and second best (very light red) results are highlighted.

> obtained using particle trajectories as supervision (see Tab. 1), demonstrating that visual supervision provides meaningful additional information for system identification [4, 8].

- *Non-Newtonian fluid*: Consistent with prior work [4, 8], this material presents the greatest challenges, and no single rendering method consistently outperforms others across all colliders.

- *Granular*: lower absolute errors are generally observed for this material, indicating relatively simpler deformation behavior compared to fluids. Despite underperforming on fluid simulations, the explicit point-based representation of 2DGS [11] achieves superior results for granular media without relying on additional dynamic reconstruction supervision, as required by MDyn-3DGS [8].

Overall, MDyn-3DGS [8] typically outperforms 2DGS [11] and DVGO [5], highlighting the advantages of incorporating dynamic reconstruction into the rendering pipeline. However, the choice of method may ultimately depend on the complexity of the specific material-collider interactions being modeled. Finally, while prior work [4, 8] has shown promising results for system identification with planar surfaces, our findings indicate that handling more complex interactions is inherently more challenging—yet essential for advancing toward models that operate under fewer assumptions.

## 6 Real-World Application: Parameter Estimation of a Dough Sample

To further validate the proposed framework, we conduct a real-world experiment using a dough sample approximately the size of a football. The dough is modeled as a non-Newtonian material due to its viscoelastic behavior, which combines solid-like elasticity with fluid-like viscosity.

**Experimental Setup.** The dough is released in free fall and collides with the edge of a box, creating a non-planar contact scenario. Upon impact, part of the dough adheres to the horizontal surface, while the rest deforms and gradually flows along the vertical edge. The experiment is recorded using five cameras placed around the scene, and in each frame the dough is segmented using SAM2 [38].

**Experiment Pipeline.** The procedure consists of three stages:

1. **Static geometry reconstruction.** We reconstruct the scene without the dough in view using 2DGS [11] and import it into AS-DiffMPM as a collider. Then, we reconstruct the initial dough geometry using the multi-view images of the first timestep. We use isotropic Gaussians to ensure an even distribution along the surface. To avoid structural collapse upon collision during the simulation, we apply the internal infilling strategy from [1]. The resulting Gaussians are then fine-tuned, with densification and pruning disabled to preserve their distribution. These adjustments lead to more realistic MPM simulations.

2. **Initial motion estimation.** We estimate the initial velocity and gravity vectors from the free-fall phase and keep them fixed in subsequent stage.

3. **System identification.** With geometry and motion initialization in place, we optimize the dough's physical parameters over the full sequence.

**Results.** The estimated physical parameters are: $\mu = 57,033.1$, $\kappa = 139,650.4$, $\tau_Y = 7,919.1$, $\eta = 25.1$. We further assess visual reconstruction quality over three seconds after collision. Tab. 3 shows the metrics averaged over 60 frames and five cameras. We observe a degradation in reconstruction quality in the later frames (see also Fig. 4). As the dough undergoes significant deformation, the Gaussians deviate from their original positions, introducing visual artifacts. A potential remedy is to incorporate *time-dependent Gaussian attributes*, which we leave for future work.

| Time after collision | PSNR ($\uparrow$) | SSIM ($\uparrow$) | LPIPS ($\downarrow$) |
|---|---|---|---|
| 1 s | $33.2 \pm 4.1$ | $0.994 \pm 0.004$ | $0.019 \pm 0.001$ |
| 2 s | $28.9 \pm 6.6$ | $0.988 \pm 0.011$ | $0.024 \pm 0.011$ |
| 3 s | $25.3 \pm 7.8$ | $0.966 \pm 0.013$ | $0.035 \pm 0.012$ |

Table 3: Evaluation of image reconstruction quality for the dough over time.

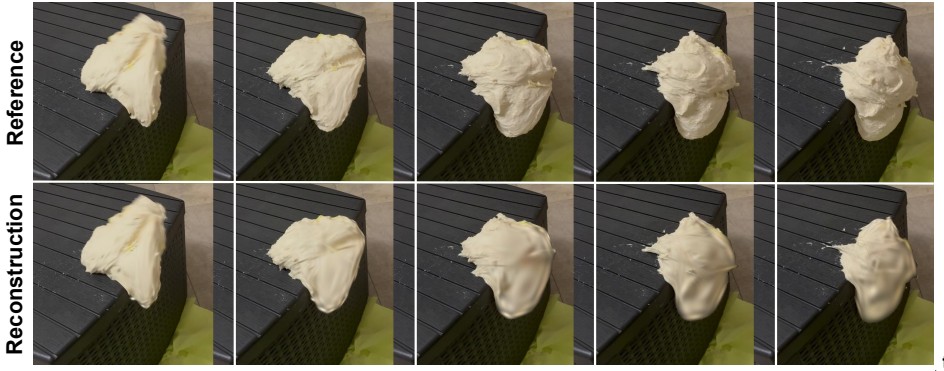

Figure 4: Example visualization of the dough experiment.

## 7 Conclusion

**Limitation.** One limitation of our work is the preliminary rigid body reconstruction-and-import into AS-DiffMPM, prior to start the system identification optimization. This restricts our approach to static colliders. Although the extension to moving rigid bodies is trivial in our framework, we leave the system identification analysis under such conditions as future work. Indeed, as significant deformation of the continuum material may occur, a carefully designed dynamic reconstruction method—similar to the one proposed in [8]—is required to mitigate visual artifacts.

In conclusion, we proposed AS-DiffMPM, a differentiable Material Point Method supporting arbitrarily shaped colliders. We validated our method quantitatively and combined it with various rendering models to perform system identification from visual observations, providing extensive results in a novel scenario where the continuum material undergoes deformations against complex colliders rather than planar surfaces.

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

# Appendix

This supplementary material provides additional results that both complement the main paper and preliminarily explore more scenarios. We invite the reader to also refer to the supplementary media attached to this paper and the project page for additional visualizations.

## A    Additional Results on System Identification

This section complements the results in Tab. 1 and Tab. 2 with additional quantitative metrics and qualitative visualizations.

### A.1    System Identification from Particle Trajectories

#### A.1.1    Quantitative Results on Geometric Error

Beyond the results reported in Tab. 1, we quantify the geometric accuracy of the simulated particle trajectories (after convergence of the physical parameter optimization) by comparing them against the reference trajectory used for supervision. Following [8, 9], we report the Chamfer Distance (CD) and Earth Mover's Distance (EMD). The Chamfer Distance is computed using the squared distance and is expressed in units of $10^3\,mm^2$. Each metric is averaged over all timesteps across the dataset. Results are presented in Tab. 4

| Material | Collider | Metrics | RP-DiffMPM | GOP-DiffMPM | AS-DiffMPM (**Ours**) |
|---|---|---|---|---|---|
| Newtonian | Box | CD $\downarrow$ | $3.903 \pm 2.603$ | $3.879 \pm 2.625$ | $3.925 \pm 2.590$ |
|  |  | EMD $\downarrow$ | $0.066 \pm 0.018$ | $0.066 \pm 0.019$ | $0.066 \pm 0.019$ |
|  | Bunny | CD $\downarrow$ | $2.948 \pm 2.060$ | $2.946 \pm 2.059$ | $2.955 \pm 2.059$ |
|  |  | EMD $\downarrow$ | $0.059 \pm 0.017$ | $0.059 \pm 0.017$ | $0.059 \pm 0.017$ |
|  | Armadillo | CD $\downarrow$ | $2.977 \pm 2.515$ | $2.964 \pm 2.512$ | $2.960 \pm 2.521$ |
|  |  | EMD $\downarrow$ | $0.064 \pm 0.025$ | $0.064 \pm 0.025$ | $0.064 \pm 0.024$ |
| Non-Newtonian | Box | CD $\downarrow$ | $16.566 \pm 6.847$ | $16.492 \pm 6.854$ | $16.558 \pm 6.868$ |
|  |  | EMD $\downarrow$ | $0.108 \pm 0.023$ | $0.107 \pm 0.023$ | $0.107 \pm 0.022$ |
|  | Bunny | CD $\downarrow$ | $12.858 \pm 5.323$ | $12.822 \pm 5.308$ | $12.826 \pm 5.353$ |
|  |  | EMD $\downarrow$ | $0.098 \pm 0.023$ | $0.098 \pm 0.023$ | $0.098 \pm 0.023$ |
|  | Armadillo | CD $\downarrow$ | $15.576 \pm 10.921$ | $15.571 \pm 10.909$ | $15.519 \pm 10.936$ |
|  |  | EMD $\downarrow$ | $0.121 \pm 0.045$ | $0.121 \pm 0.045$ | $0.121 \pm 0.045$ |
| Granular | Box | CD $\downarrow$ | $3.478 \pm 1.054$ | $3.485 \pm 1.090$ | $3.448 \pm 1.043$ |
|  |  | EMD $\downarrow$ | $0.064 \pm 0.005$ | $0.064 \pm 0.006$ | $0.063 \pm 0.005$ |
|  | Bunny | CD $\downarrow$ | $3.084 \pm 0.820$ | $3.067 \pm 0.847$ | $3.055 \pm 0.851$ |
|  |  | EMD $\downarrow$ | $0.058 \pm 0.006$ | $0.058 \pm 0.005$ | $0.058 \pm 0.005$ |
|  | Armadillo | CD $\downarrow$ | $2.896 \pm 0.857$ | $2.860 \pm 0.845$ | $2.858 \pm 0.902$ |
|  |  | EMD $\downarrow$ | $0.056 \pm 0.006$ | $0.056 \pm 0.006$ | $0.056 \pm 0.006$ |

Table 4: Evaluation of geometric error using CD and EMD for different materials and colliders.

### A.1.2 Qualitative Visualizations

We provide example visualizations of the 3D particle trajectories used in our experiments (see Fig. 5). Please refer to the supplementary material attached to this paper for more videos.

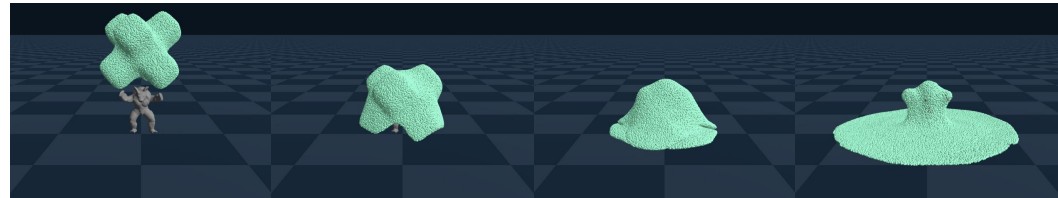

Figure 5: Particle trajectories of a Newtonian material colliding with the Armadillo rigid body.

## A.2 System Identification from Visual Observations

### A.2.1 Quantitative Results on Image Reconstruction

In addition to the results shown in Tab. 2, we assess the image reconstruction quality of all models after convergence of the physical parameter optimization. We report in Tab. 5 the standard metrics used in radiance field evaluation: Peak Signal-to-Noise Ratio (PSNR), Structural Similarity Index Measure (SSIM), and Learned Perceptual Image Patch Similarity (LPIPS). Each metric is averaged over all views and timesteps across the dataset.

| Material | Collider | Metrics | DVGO [5] | 2DGS [11] | MDyn-3DGS [8] |
|---|---|---|---|---|---|
| Newtonian | Box | PSNR ↑ | $35.318 \pm 4.901$ | $35.669 \pm 6.779$ | $41.300 \pm 2.130$ |
| | | SSIM ↑ | $0.985 \pm 0.010$ | $0.978 \pm 0.022$ | $0.997 \pm 0.002$ |
| | | LPIPS ↓ | $0.023 \pm 0.016$ | $0.022 \pm 0.019$ | $0.017 \pm 0.007$ |
| | Bunny | PSNR ↑ | $36.491 \pm 4.217$ | $36.949 \pm 6.083$ | $41.537 \pm 1.856$ |
| | | SSIM ↑ | $0.987 \pm 0.009$ | $0.981 \pm 0.019$ | $0.998 \pm 0.001$ |
| | | LPIPS ↓ | $0.021 \pm 0.014$ | $0.019 \pm 0.017$ | $0.015 \pm 0.006$ |
| | Armadillo | PSNR ↑ | $36.905 \pm 3.409$ | $37.484 \pm 5.340$ | $41.888 \pm 1.395$ |
| | | SSIM ↑ | $0.987 \pm 0.008$ | $0.982 \pm 0.017$ | $0.998 \pm 0.001$ |
| | | LPIPS ↓ | $0.021 \pm 0.012$ | $0.019 \pm 0.015$ | $0.015 \pm 0.005$ |
| Non-Newtonian | Box | PSNR ↑ | $28.331 \pm 5.462$ | $29.468 \pm 6.658$ | $48.521 \pm 2.291$ |
| | | SSIM ↑ | $0.973 \pm 0.014$ | $0.980 \pm 0.015$ | $0.998 \pm 0.001$ |
| | | LPIPS ↓ | $0.037 \pm 0.014$ | $0.023 \pm 0.015$ | $0.011 \pm 0.003$ |
| | Bunny | PSNR ↑ | $28.871 \pm 5.077$ | $31.718 \pm 5.184$ | $48.659 \pm 2.278$ |
| | | SSIM ↑ | $0.975 \pm 0.012$ | $0.986 \pm 0.011$ | $0.998 \pm 0.001$ |
| | | LPIPS ↓ | $0.035 \pm 0.012$ | $0.018 \pm 0.011$ | $0.010 \pm 0.003$ |
| | Armadillo | PSNR ↑ | $27.196 \pm 5.573$ | $32.176 \pm 4.459$ | $48.762 \pm 1.814$ |
| | | SSIM ↑ | $0.971 \pm 0.013$ | $0.986 \pm 0.009$ | $0.998 \pm 0.001$ |
| | | LPIPS ↓ | $0.038 \pm 0.014$ | $0.018 \pm 0.010$ | $0.010 \pm 0.002$ |
| Granular | Box | PSNR ↑ | $31.338 \pm 5.558$ | $31.055 \pm 7.419$ | $40.527 \pm 3.951$ |
| | | SSIM ↑ | $0.967 \pm 0.023$ | $0.957 \pm 0.040$ | $0.992 \pm 0.006$ |
| | | LPIPS ↓ | $0.046 \pm 0.028$ | $0.035 \pm 0.029$ | $0.022 \pm 0.009$ |
| | Bunny | PSNR ↑ | $32.487 \pm 4.841$ | $32.063 \pm 6.870$ | $42.522 \pm 2.127$ |
| | | SSIM ↑ | $0.972 \pm 0.018$ | $0.963 \pm 0.034$ | $0.994 \pm 0.003$ |
| | | LPIPS ↓ | $0.042 \pm 0.024$ | $0.031 \pm 0.026$ | $0.018 \pm 0.007$ |
| | Armadillo | PSNR ↑ | $32.204 \pm 4.577$ | $31.361 \pm 6.936$ | $41.900 \pm 2.122$ |
| | | SSIM ↑ | $0.970 \pm 0.018$ | $0.959 \pm 0.035$ | $0.993 \pm 0.003$ |
| | | LPIPS ↓ | $0.044 \pm 0.024$ | $0.034 \pm 0.027$ | $0.021 \pm 0.008$ |

Table 5: Evaluation of image reconstruction using PSNR, SSIM, and LPIPS for different materials and colliders.

### A.2.2 Qualitative Visualizations

We provide example visualizations of the models at convergence (see Fig. 6 and Fig. 7). We remind that, prior to the physical parameters optimization, the rigid collider is imported into AS-DiffMPM for collision handling and the continuum material is segmented out from the ground truth views. During physical parameters optimization, the visual observations containing the segmented-out continuum object are used for supervision. Please refer to the supplementary material attached to this paper for videos, along with the reconstruction metrics and estimated physical parameters.

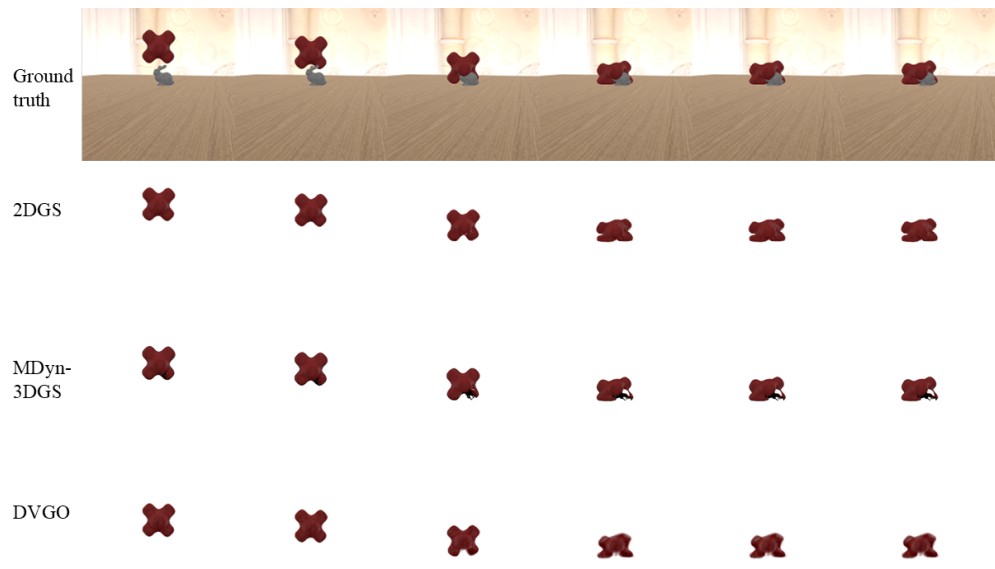

Figure 6: Non-Newtonian material colliding with the Bunny rigid body.

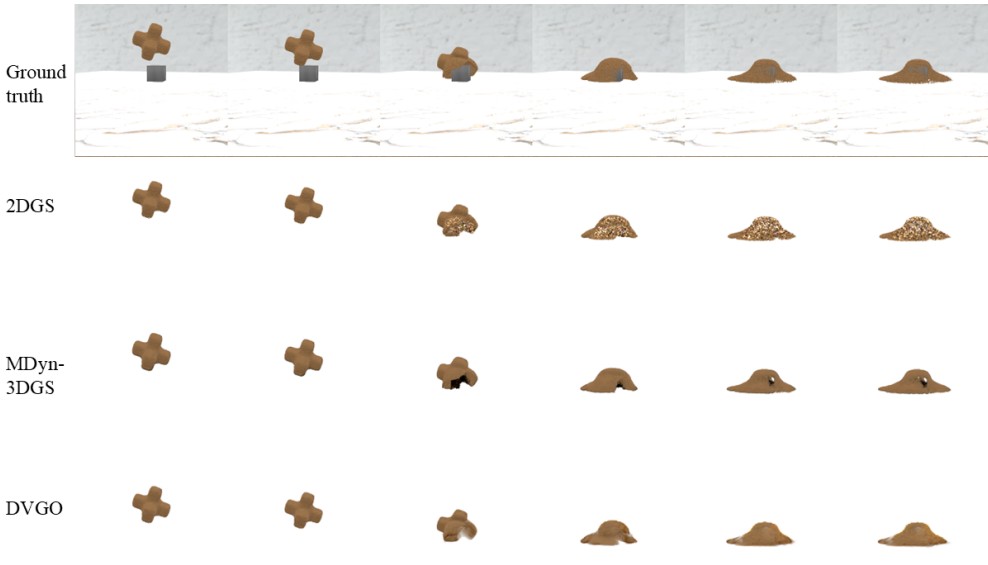

Figure 7: Sand material colliding with the Box rigid body.

# B    Mesh vs. 2DGS as Rigid Body Collider

In contrast to the experiments presented in the main paper (Sec.5), where the collider imported into AS-DiffMPM is represented as a mesh, in this analysis we segment and reconstruct the collider using 2DGS[11], and import it into AS-DiffMPM as described in Sec. 4.2. We then perform the same system identification experiment based on particle trajectories to evaluate the impact of using a mesh compared to 2DGS as rigid body collider representation. Results are presented in Tab. 6 and Tab. 7. As expected, 2DGS-based colliders obtains lower accuracy. This is due to the reconstructed Gaussians not being as precise as a mesh, which introduce noise into the Collision grid (Sec. 4.1.1) and result in less accurate collision handling. However, we remark that our AS-DiffMPM is versatile enough to support both mesh and 2DGS-based colliders. This flexibility is particularly valuable for real-world scenarios, where colliders may need to be reconstructed on-the-fly using 2DGS to enable simulation.

| Material | Collider | Parameters | AS-DiffMPM w/ 2DGS | AS-DiffMPM w/ Mesh |
|---|---|---|---|---|
| Newtonian | Box | $\log_{10}(\mu)$ | **4.36 $\pm$ 2.66** | 5.08 $\pm$ 2.73 |
| | | $\log_{10}(\kappa)$ | 36.26 $\pm$ 24.63 | **27.81 $\pm$ 33.49** |
| | Bunny | $\log_{10}(\mu)$ | 7.03 $\pm$ 4.09 | **4.35 $\pm$ 5.24** |
| | | $\log_{10}(\kappa)$ | 81.30 $\pm$ 31.53 | **31.37 $\pm$ 50.02** |
| | Armadillo | $\log_{10}(\mu)$ | 8.75 $\pm$ 6.17 | **0.4 $\pm$ 0.76** |
| | | $\log_{10}(\kappa)$ | 86.74 $\pm$ 47.68 | **0.22 $\pm$ 0.24** |
| Non-Newtonian | Box | $\log_{10}(\mu)$ | 55.51 $\pm$ 42.28 | **23.51 $\pm$ 30.07** |
| | | $\log_{10}(\kappa)$ | 126.10 $\pm$ 99.60 | **27.06 $\pm$ 19.12** |
| | | $\log_{10}(\tau_Y)$ | 11.18 $\pm$ 8.63 | **7.28 $\pm$ 3.75** |
| | | $\log_{10}(\eta)$ | 56.71 $\pm$ 35.83 | **17.7 $\pm$ 15.17** |
| | Bunny | $\log_{10}(\mu)$ | 53.12 $\pm$ 44.90 | **21.92 $\pm$ 36.89** |
| | | $\log_{10}(\kappa)$ | 110.70 $\pm$ 73.58 | **26.77 $\pm$ 25.09** |
| | | $\log_{10}(\tau_Y)$ | 16.40 $\pm$ 5.74 | **2.48 $\pm$ 1.87** |
| | | $\log_{10}(\eta)$ | 56.57 $\pm$ 35.17 | **39.1 $\pm$ 36.2** |
| | Armadillo | $\log_{10}(\mu)$ | 25.22 $\pm$ 24.02 | **7.14 $\pm$ 6.64** |
| | | $\log_{10}(\kappa)$ | 124.88 $\pm$ 75.89 | **56.26 $\pm$ 76.97** |
| | | $\log_{10}(\tau_Y)$ | 28.43 $\pm$ 19.45 | **5.07 $\pm$ 5.79** |
| | | $\log_{10}(\eta)$ | 51.76 $\pm$ 30.75 | **44.98 $\pm$ 41.34** |
| Granular | Box | $\theta_{fric}$ | 1.86 $\pm$ 1.14 | **1.22 $\pm$ 0.71** |
| | Bunny | $\theta_{fric}$ | 0.91 $\pm$ 1.15 | **0.06 $\pm$ 0.07** |
| | Armadillo | $\theta_{fric}$ | 1.38 $\pm$ 1.08 | **0.51 $\pm$ 0.96** |

Table 6: **System identification from particle trajectories.** Evaluation (reported as absolute error) of physical parameter estimation when using mesh and 2DGS [11] as collider.

| Material | Collider | Metrics | AS-DiffMPM w/ 2DGS | AS-DiffMPM w/ Mesh |
|---|---|---|---|---|
| Newtonian | Box | CD ↓ | **3.893 ± 2.586** | 3.925 ± 2.590 |
| | | EMD ↓ | 0.066 ± 0.018 | 0.066 ± 0.019 |
| | Bunny | CD ↓ | 2.955 ± 2.049 | **2.937 ± 2.059** |
| | | EMD ↓ | 0.059 ± 0.017 | 0.059 ± 0.017 |
| | Armadillo | CD ↓ | 2.962 ± 2.515 | **2.960 ± 2.521** |
| | | EMD ↓ | 0.064 ± 0.024 | 0.064 ± 0.024 |
| Non-Newtonian | Box | CD ↓ | 16.629 ± 6.882 | **16.558 ± 6.868** |
| | | EMD ↓ | 0.108 ± 0.023 | **0.107 ± 0.022** |
| | Bunny | CD ↓ | 12.867 ± 5.355 | **12.826 ± 5.353** |
| | | EMD ↓ | 0.098 ± 0.023 | 0.098 ± 0.023 |
| | Armadillo | CD ↓ | 15.538 ± 10.854 | **15.519 ± 10.936** |
| | | EMD ↓ | 0.121 ± 0.045 | 0.121 ± 0.045 |
| Granular | Box | CD ↓ | 3.468 ± 1.042 | **3.448 ± 1.043** |
| | | EMD ↓ | 0.064 ± 0.005 | **0.063 ± 0.005** |
| | Bunny | CD ↓ | 3.094 ± 0.867 | **3.055 ± 0.851** |
| | | EMD ↓ | **0.057 ± 0.005** | 0.058 ± 0.005 |
| | Armadillo | CD ↓ | 2.872 ± 0.886 | **2.858 ± 0.902** |
| | | EMD ↓ | 0.056 ± 0.006 | 0.056 ± 0.006 |

Table 7: **System identification from particle trajectories.** Evaluation of geometric error when using mesh or 2DGS as collider.

## C   Exploring More Conditions for System Identification

We further benchmark our AS-DiffMPM framework under more conditions, both in terms of colliders and materials. This section is not intended to provide an in-depth analysis for each condition, but rather to explore a wide range of possibilities, aiming to possibly identify which ones deserve more thorough analysis in future work. All experiments are conducted using the pipeline described in Sec. 5.1 for system identification from particle trajectories.

### C.1   Same Colliders in Different Conditions

We use the three colliders (Box, Bunny and Armadillo) of our analysis in the main paper but changing characteristics. For each case, results are reported per material type, averaging over the colliders (see Tab. 8)

- **Condition 1.** The collider is moved to the side (instead of being underneath the object) such that the continuum material first impacts the ground and then interacts with the collider as it spreads. We perform nine experiments—comprising three material types and three colliders.

- **Condition 2.** Each collider is tested at two scales (0.5 and 1.5), for all material types and colliders, resulting in 18 experiments.

- **Condition 3.** We combine all the colliders at once, placing them side by side, so that the material interacts with all of them upon collision. This results in three experiments, one per material type

Overall, Condition 1 appears to be the one with more significant challenges in physical parameter estimation, likely due to this ground-first-collider-next complex interactions between the material and the collider.

| Material | Parameter | Cond. 1 | Cond. 2 | Cond. 3 |
|---|---|---|---|---|
| Newtonian | $\log_{10}(\mu)$ | $10.51 \pm 12.09$ | $5.21 \pm 7.02$ | $6.85$ |
|  | $\log_{10}(\kappa)$ | $39.82 \pm 9.65$ | $26.88 \pm 25.91$ | $47.34$ |
| Non-Newtonian | $\log_{10}(\mu)$ | $23.02 \pm 10.31$ | $22.17 \pm 12.81$ | $3.41$ |
|  | $\log_{10}(\kappa)$ | $50.34 \pm 7.78$ | $23.15 \pm 15.64$ | $41.98$ |
|  | $\log_{10}(\tau_Y)$ | $9.56 \pm 5.90$ | $5.34 \pm 4.26$ | $7.59$ |
|  | $\log_{10}(\eta)$ | $34.41 \pm 12.74$ | $21.03 \pm 7.98$ | $32.02$ |
| Granular | $\theta_{fric}$ | $4.02 \pm 1.70$ | $1.09 \pm 0.93$ | $3.04$ |

Table 8: System identification errors across three conditions.

| Material | Parameter | S-MPM | RP-DiffMPM | GOP-DiffMPM | AS-DiffMPM (**Ours**) |
|---|---|---|---|---|---|
| Newtonian | $\log_{10}(\mu)$ | 7.51 | 8.03 | 7.63 | 7.98 |
|  | $\log_{10}(\kappa)$ | 31.98 | 35.21 | 32.45 | 33.21 |
| Non-Newtonian | $\log_{10}(\mu)$ | 24.09 | 24.57 | 24.91 | 24.03 |
|  | $\log_{10}(\kappa)$ | 19.01 | 21.12 | 20.19 | 19.03 |
|  | $\log_{10}(\tau_Y)$ | 4.91 | 5.12 | 4.78 | 4.54 |
|  | $\log_{10}(\eta)$ | 30.53 | 32.10 | 31.05 | 31.21 |
| Granular | $\theta_{fric}$ | 0.15 | 0.56 | 0.22 | 0.12 |

Table 9: System identification errors for planar collider.

## C.2 More Colliders and Material Types

In this section, we focus both on different colliders and material types, comparing our AS-DiffMPM framework with baselines. For the former, we first explore a comparison using the condition adopted in previous works [4, 7, 8, 9], i.e., a planar collider, and then with a bowl-shaped collider. Instead, for the latter, we explore both further material types and shapes.

**Planar Collider.** We conduct experiments comparing our AS-DiffMPM with both the baselines used in our main paper (GOP-DiffMPM and RP-DiffMPM) and the Standard MPM (S-MPM) collision strategy adopted by previous works [4, 7, 8, 9]. In S-MPM, collisions are resolved at the Grid Operations stage (Sec. 3.1): for a planar surface defined by point $\mathbf{x}_b$ and normal $\mathbf{n}_b$, the nodal velocity $\mathbf{v}_g$ is set to zero for all grid nodes $\mathbf{x}_g$ such that $(\mathbf{x}_g - \mathbf{x}_b) \cdot \mathbf{n}_b \leq 0$. In contrast, AS-DiffMPM represents the same planar collider as a mesh surface positioned at $\mathbf{x}_b$, and applies particle-wise collision handling during both P2G and G2P steps. We performed three experiments, one per material type (see Tab. 9). The S-MPM achieves slightly lower errors for some parameters, likely due to its use of analytically defined boundary conditions, which may offer improved numerical stability in simple geometries.

**Bowl Collider.** We import a bowl-shaped collider into our framework and place it such that the falling material impacts its open surface—causing some particles to enter the bowl while others fall outside. We conducted three experiments, one per material type (see Tab. 10). The general trend observed in the main paper appears to be consistent here, with our method outperforming the baselines.

**More material types.** While the main focus of our work is on Newtonian, non-Newtonian and granular materials, we explore here also with elastic and plasticine materials. For elastic material, we consider Young's modulus ($E$) and Poisson's ration ($\nu$), while for plasticine material, we consider also yield stress ($\tau_Y$). We refer the reader to [4] for the constitutive models. Each method was evaluated on six experiments: two material types and with three colliders. The results are presented in Tab. 11, averaged over the colliders. Interestingly, all methods generally perform well on the elastic material. This may be due to the simpler nature of the interaction between the elastic material and the collider—since the material tends to bounce away upon contact, the particle-wise collision handling mechanism of AS-DiffMPM has less impact. Indeed, in these scenarios, no fine-grained interaction (e.g., material flowing around sharp corners) occurs.

| Material | Parameter | RP-DiffMPM | GOP-DiffMPM | AS-DiffMPM (**Ours**) |
|---|---|---|---|---|
| Newtonian | $\log_{10}(\mu)$ | 7.84 | 7.29 | 6.01 |
| | $\log_{10}(\kappa)$ | 30.12 | 28.45 | 25.67 |
| Non-Newtonian | $\log_{10}(\mu)$ | 19.31 | 20.05 | 15.03 |
| | $\log_{10}(\kappa)$ | 38.22 | 35.79 | 29.84 |
| | $\log_{10}(\tau_Y)$ | 7.31 | 6.98 | 5.15 |
| | $\log_{10}(\eta)$ | 26.78 | 28.45 | 20.94 |
| Granular | $\theta_{\mathrm{fric}}$ | 3.01 | 1.17 | 1.41 |

Table 10: System identification errors for bowl collider.

| Material | Parameter | RP-DiffMPM | GOP-DiffMPM | AS-DiffMPM (**Ours**) |
|---|---|---|---|---|
| Elastic | $\log_{10}(E)$ | $2.01 \pm 0.45$ | $2.87 \pm 1.18$ | $1.91 \pm 0.96$ |
| | $\nu$ | $0.36 \pm 0.43$ | $1.23 \pm 0.32$ | $0.49 \pm 0.12$ |
| Plasticine | $\log_{10}(E)$ | $28.45 \pm 3.12$ | $16.88 \pm 1.84$ | $12.07 \pm 2.21$ |
| | $\nu$ | $9.29 \pm 5.04$ | $10.24 \pm 3.03$ | $8.21 \pm 1.02$ |
| | $\log_{10}(\tau_Y)$ | $9.44 \pm 2.05$ | $8.76 \pm 1.88$ | $6.23 \pm 1.47$ |

Table 11: System identification errors for elastic and plasticine material types.

| Shape | Material | Parameter | RP-DiffMPM | GOP-DiffMPM | AS-DiffMPM (**Ours**) |
|---|---|---|---|---|---|
| Droplet | Newtonian | $\log_{10}(\mu)$ | $7.81 \pm 3.52$ | $6.94 \pm 4.41$ | $5.87 \pm 3.37$ |
| | | $\log_{10}(\kappa)$ | $32.33 \pm 3.25$ | $21.91 \pm 8.13$ | $24.50 \pm 3.98$ |
| Letter | Newtonian | $\log_{10}(\mu)$ | $8.12 \pm 3.49$ | $7.22 \pm 2.38$ | $6.05 \pm 1.35$ |
| | | $\log_{10}(\kappa)$ | $31.01 \pm 5.40$ | $23.12 \pm 2.17$ | $22.36 \pm 6.04$ |
| Cream | Non-Newtonian | $\log_{10}(\mu)$ | $19.42 \pm 7.61$ | $18.67 \pm 2.53$ | $14.73 \pm 5.12$ |
| | | $\log_{10}(\kappa)$ | $46.50 \pm 12.09$ | $44.12 \pm 3.88$ | $38.08 \pm 4.65$ |
| | | $\log_{10}(\tau_Y)$ | $6.92 \pm 0.58$ | $6.58 \pm 0.47$ | $4.97 \pm 0.34$ |
| | | $\log_{10}(\eta)$ | $54.88 \pm 12.72$ | $46.41 \pm 5.85$ | $40.01 \pm 7.43$ |
| Toothpaste | Non-Newtonian | $\log_{10}(\mu)$ | $20.05 \pm 5.45$ | $19.18 \pm 1.29$ | $15.30 \pm 1.06$ |
| | | $\log_{10}(\kappa)$ | $44.44 \pm 4.33$ | $44.99 \pm 3.04$ | $39.72 \pm 3.89$ |
| | | $\log_{10}(\tau_Y)$ | $7.35 \pm 1.61$ | $6.97 \pm 0.52$ | $5.12 \pm 0.38$ |
| | | $\log_{10}(\eta)$ | $62.19 \pm 19.06$ | $47.55 \pm 21.95$ | $41.17 \pm 9.01$ |
| Torus | Elastic | $\log_{10}(E)$ | $1.85 \pm 0.83$ | $2.87 \pm 0.75$ | $2.69 \pm 0.62$ |
| | | $\nu$ | $0.22 \pm 0.02$ | $0.25 \pm 0.02$ | $0.19 \pm 0.01$ |
| Bird | Elastic | $\log_{10}(E)$ | $2.31 \pm 0.90$ | $2.94 \pm 0.81$ | $2.51 \pm 0.69$ |
| | | $\nu$ | $0.22 \pm 0.02$ | $0.26 \pm 0.02$ | $0.21 \pm 0.01$ |
| Playdoh | Plasticine | $\log_{10}(E)$ | $17.38 \pm 1.14$ | $15.89 \pm 1.03$ | $11.62 \pm 0.89$ |
| | | $\nu$ | $8.31 \pm 3.03$ | $9.27 \pm 2.02$ | $6.22 \pm 2.01$ |
| | | $\log_{10}(\tau_Y)$ | $9.91 \pm 1.46$ | $8.70 \pm 0.62$ | $6.74 \pm 0.51$ |
| Cat | Plasticine | $\log_{10}(E)$ | $18.11 \pm 1.20$ | $16.32 \pm 1.05$ | $12.01 \pm 0.95$ |
| | | $\nu$ | $8.30 \pm 2.71$ | $8.28 \pm 1.02$ | $6.23 \pm 1.98$ |
| | | $\log_{10}(\tau_Y)$ | $10.02 \pm 0.84$ | $9.14 \pm 0.71$ | $7.20 \pm 0.57$ |
| Trophy | Granular | $\theta_{\mathrm{fric}}$ | $2.19 \pm 0.15$ | $1.95 \pm 0.12$ | $1.24 \pm 0.09$ |

Table 12: System identification errors for different material shapes.

**More material shapes.** We use the 9 shapes and their associated material types introduced in [4]. Each method is tested on all 9 shapes, with each shape colliding against the three colliders, resulting in 27 experiments per method. The results are presented in Tab. 12, averaged over the colliders.

# D    Extensions of AS-DiffMPM

Due to our collision handling mechanism integrated into the *P2G* and *G2P* stages, the AS-DiffMPM framework can accommodate additional physical constraints. In the following, we present two illustrative examples.

## D.1    Material Cutting

We consider a thin surface intersecting with a continuum material. We implement a *separating surface* behavior where we adjust the velocity of the particle $\mathbf{v}_p$ based on its relative position to the surface. Concretely, during the *G2P* stage, for each incompatible particle (Sec. 4.1.2), we compute the *signed* particle normal $\bar{\mathbf{n}}_p = \mathbf{n}_p T_p$, and update the velocity according to the particle's motion relative to the surface. If the particle is moving away from the boundary (i.e., $\mathbf{v}_p \cdot \bar{\mathbf{n}}_p > 0$), its velocity remains unchanged. Otherwise, we project the velocity along the surface: $\mathbf{v}_p^{proj} = \mathbf{v}_p - (\mathbf{v}_p \cdot \bar{\mathbf{n}}_p)\,\bar{\mathbf{n}}_p$.

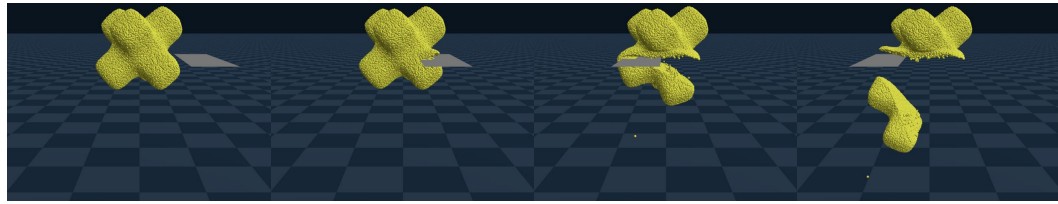

Figure 8: A thin surface cutting an elastic object.

## D.2    Collider-to-Continuum Coupling

We present a scenario where a moving collider can interact with the continuum material. During the *G2P* stage, for each incompatible particle, we correct the velocity as $\bar{\mathbf{v}}_p = \mathbf{v}_p + \mathbf{v}_r$, where $\mathbf{v}_r$ is the rigid body velocity at position $\mathbf{x}_p$.

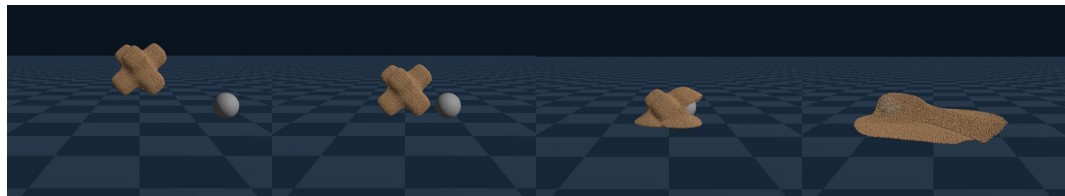

Figure 9: A ball colliding with a sand object.

# E    Values of Physical Parameters for Ground-truth Rollouts

For a fair comparison, we generate ground-truth simulation rollouts using the same physical parameter values as in [4], listed in Tab. 13 for convenience. Each ground-truth rollout is executed three times, one for each collider, totaling 75 rollouts. We remark that for each rollout a separate training run of system identification is performed, where an initial guess of the physical parameters is optimized by comparing the current rollout against the reference (either in the form of particle trajectories or visual observations). For each material type, we use the same initial guess across all training runs.

| Newtonian (Initial Guess: $\mu = 10$, $\kappa = 10^4$) | | | | | | | | | |
|---|---|---|---|---|---|---|---|---|---|
| Parameter | 1 | 2 | 3 | 4 | 5 | 6 | 7 | 8 | 9 | 10 |
| $\mu$ | 19.46 | 436.62 | 155.83 | 121.76 | 49.09 | 38.44 | 64.16 | 228.71 | 552.98 | 106.93 |
| $\kappa$ | 56075.55 | 152696.25 | 193525.59 | 257356.05 | 518012.47 | 13772.52 | 358237.13 | 11041.06 | 16789.77 | 112569.73 |

| Non-Newtonian (Initial Guess: $\mu = 100$, $\kappa = 10^5$, $\tau_Y = 10$, $\eta = 1$) | | | | | | | | | |
|---|---|---|---|---|---|---|---|---|---|
| Parameter | 1 | 2 | 3 | 4 | 5 | 6 | 7 | 8 | 9 | 10 |
| $\mu$ | 13209.25 | 65351.08 | 43757.04 | 36027.61 | 19593.71 | 20522.72 | 51549.45 | 121865.90 | 241579.97 | 33764.59 |
| $\kappa$ | 201566.59 | 171054.03 | 249639.94 | 134751.55 | 121836.33 | 14494.30 | 370317.66 | 32859.59 | 30324.98 | 122896.10 |
| $\tau_Y$ | 1151.42 | 7491.70 | 3964.94 | 5061.12 | 1462.78 | 4153.38 | 3203.67 | 1192.76 | 1251.29 | 4689.16 |
| $\eta$ | 6.68 | 26.69 | 23.27 | 22.31 | 38.83 | 27.24 | 20.43 | 10.27 | 10.62 | 22.89 |

| Granular (Initial Guess: $\theta_{fric} = 10$) | | | | |
|---|---|---|---|---|
| Parameter | 1 | 2 | 3 | 4 | 5 |
| $\theta_{fric}$ | 30.6577 | 32.3751 | 26.8816 | 29.3458 | 42.2861 |

Table 13: Values of physical parameters for ground-truth simulation rollouts.

# F  Implementation Details

We implemented our framework using Python and Taichi for differentiable programming and parallel computation. AS-DiffMPM is built upon the open source DiffMPM implementation in [4] and subsequent works [7, 8]. We run the experiments on NVIDIA RTX 3080 and 4090 graphics cards.

