# OpenReview forum: "Gaussian-Augmented Physics Simulation and System Identification with Complex Colliders"
_NeurIPS.cc/2025/Conference — NeurIPS 2025 poster_

### Official Review · Reviewer_CpqV · 2025-06-21

**Clarity:** 3
**Significance:** 2
**Originality:** 2
**Rating:** 4
**Confidence:** 4

**Summary:**

This paper addresses the challenge of system identification using arbitrarily shaped colliders. To achieve this, it enhances an existing differentiable Material Point Method (MPM) by incorporating a differentiable collision handling mechanism. This addition enables the estimation of physical properties using differentiable renderers and simulators. Experiments conducted on several synthetic benchmarks demonstrated the effectiveness of the proposed approach.

**Questions:**

1.	Per Weakness 1, could the authors present more results, including but not limited to the following scenarios: (1) move the collider to a corner of the floor instead of directly below the object, (2) make the size of the collider (i.e., the bounding-box size) much greater than that of the object, (3) use an open-surface collider, e.g., a bowl-shaped mesh or a cylinder open at the ends, and (4) place multiple colliders side by side?

2.	Per Weakness 2, could the authors present results of system identification on elastic and plastic objects?

3.	Per Weakness 3, could the authors conduct at least one real-world experiment to verify its practical usefulness?

4.	How long does it take (on average) to finish an identification task for the proposed method and the baseline approaches?

**Ethical Concerns:**

["NO or VERY MINOR ethics concerns only"]

**Final Justification:**

The authors' response addressed my initial concerns about the limited evaluation and lack of real-world experiments. The additional results enhance the paper's completeness and demonstrate the potential practicality of the proposed method. Therefore, I decided to revise my rating to 4.

**Limitations:**

The authors list several limitations in the paper, including the requirement for rigid-body reconstruction and the use of static colliders.

The objects considered in the paper appear to have only one shape (i.e., the cross shape) and uniform material properties. It would be interesting to have some results on objects with other shapes and heterogeneous material compositions.

Another interesting experiment would be to replace the colliders of rigid objects with those of deformable objects.

**Paper Formatting Concerns:**

No paper formatting concerns.

**Quality:**

3

**Strengths And Weaknesses:**

**Strengths**:

1. The paper is well written and easy to follow.

2. It extends the existing differentiable MPM simulator with support for a differentiable collision handling strategy that accommodates arbitrarily shaped colliders.

3. The proposed collision handling strategy is generalized to primitives with associated normals, e.g., 2D Gaussian disks.

**Weaknesses**:

1. The proposed method aims to address system identification with complex colliders. While it did present results of non-planar colliders, it still lacks *variety* and *complexity* regarding the (a) location, (b) size, (c) shape, and (d) number of colliders in the considered scenarios.

2. The experiments only show results of a limited range of material types, i.e., only consider fluids and granular media. However, previous work [1] also considers elastic and plastic materials for experiments. This observation may indicate *a lack of generality* of the proposed framework.

3. The method claims that an ad hoc solution for system identification with complex colliders is necessary, yet it only presents results based on synthetic data. In a synthetic environment, we can actually estimate the material properties of objects by using a planar collider to collect simulation trajectories (since we have full control of the simulation). Consequently, a more practical application of the proposed method should involve using multi-view videos captured from *real-world environments*, which show objects colliding with complex colliders. However, the method lacks results that reflect this practical scenario.

---

> ### Author Rebuttal · Authors · 2025-07-31
>
> We thank the reviewer for their valuable feedback. Specifically, we appreciated the comments on quality, clarity and recognizing the value of handling different primitives (i.e., mesh and 2D Gaussians) in our framework.
>
> ***In this rebuttal, we provide additional experiments on various aspect of the colliders, evaluate our method on additional material types and shapes, and provide additional details on the training time.***
>
> Below, we address every point raised by the reviewer.
>
> > [Q1]. Per Weakness 1, could the authors present more results [...]
>
> ## (1) Move collider to the side
> We conducted additional experiments in which the collider is moved to the side, so that the material **first impacts the ground** and **then interacts with the collider** as it spreads along the surface. We evaluated our AS-DiffMPM across nine experiments—comprising three material types and three colliders. For brevity, we report the results per material type, averaged over the three colliders (Box, Bunny and Armadillo)
>
> This configuration appears to introduce significant challenges in physical parameter estimation, likely due to this *ground-first-collider-next* complex interactions between the material and the collider. However, a more extensive evaluation would be needed to draw definitive conclusions.
>
> ### Newtonian
> | Parameter | Error |
> | --------- | ----- |
> | μ     |  10.51  ± 12.09   |
> | κ  | 39.82  ± 9.65   |
>
>
> ### Non-Newtonian
> | Parameter              | Error |
> | ---------------------- | ----- |
> | μ                  | 23.02  ±   10.31   |
> | κ               | 50.34  ± 7.78  |
> |  τY | 9.56  ± 5.90   |
> | η                 | 34.41  ±  12.74  |
>
>
> ### Granular
> | Parameter                         | Error |
> | --------------------------------- | ----- |
> | θ_fric | 4.02  ±  1.7  |
>
>
> ## (2) Colliders of different size
> For this analysis, we used AS-DiffMPM and ran 18 experiments: three material types with three colliders, each tested at two scales (0.5 and 1.5), resulting in 3 × 3 × 2 = 18 experiments. The results are averaged over the colliders.
>
> No significant differences were observed compared to the results in Table 1 of our manuscript, although a more extensive evaluation would be needed to confirm this observation.
>
> ### Newtonian
> | Parameter | Error |
> | --------- | ----- |
> | μ     |  	5.21 ± 7.02   |
> | κ  | 26.88 ± 25.91  |
>
>
> ### Non-Newtonian
> | Parameter              | Error |
> | ---------------------- | ----- |
> | μ                  | 22.17 ± 12.81   |
> | κ               | 23.15 ± 15.64  |
> |  τY | 	5.34 ± 4.26   |
> | η                 | 21.03 ± 7.98  |
>
>
> ### Granular
> | Parameter                         | Error |
> | --------------------------------- | ----- |
> | θ_fric | 1.09 ± 0.93 |
>
>
> ## (3) Bowl collider
> We imported a bowl-shaped collider into our framework and conducted a comparative study of our AS-DiffMPM method against the baselines. The bowl was positioned such that the falling material impacts its open surface—causing some particles to enter the bowl while others fall outside of it. We conducted three experiments, one for each material type. The results are reported below.
>
> | Material      | Parameter        | RP-DiffMPM | GOP-DiffMPM | AS-DiffMPM  |
> | ------------- | ---------------- | ---------- | ----------- | ----------------- |
> | Newtonian     | μ                | 7.84       | 7.29        | **6.01**              |
> |               | κ                | 30.12      | 28.45       | **25.67**             |
> | Non-Newtonian | μ                | 19.31      | 20.05       | **15.03**             |
> |               | κ                | 38.22      | 35.79       | **29.84**             |
> |               | τY    | 7.31       | 6.98        | **5.15**              |
> |               | η                | 26.78      | 28.45       | **20.94**             |
> | Granular      | θ_fric | 3.01       | **1.17**        | 1.41              |
>
>
> ## (4) Multiple colliders
>
> We conduct three experiments—one for each material type—in scenes containing multiple colliders. Specifically, we place the Box, Bunny, and Armadillo colliders side by side and below the falling material, so that it interacts with all three as it spreads. These experiments are performed using our AS-DiffMPM method. The results are presented below.
>
> | Material          | Parameter        | Error        |
> | ----------------- | ---------------- | ------------ |
> | **Newtonian**     | μ                | 6.85   |
> |                   | κ                | 47.34  |
> | **Non-Newtonian** | μ                | 3.41 |
> |                   | κ                | 41.98 |
> |                   | τY               | 7.59  |
> |                   | η                | 32.02 |
> | **Granular**      | θ_fric            | 3.04         |
>
>
>
> ## (5) More material shapes
> We thank the reviewer for the constructive feedback in the limitations section. We agree that evaluating system identification on a wider variety of material shapes would strengthen our empirical analysis. To this end, we use 9 shapes and their associated material types from previous work [1]. We evaluate our AS-DiffMPM method and the baselines. Each method is tested on all 9 shapes, with each shape colliding against the three colliders, resulting in 27 experiments per method. The results, averaged over the colliders, are presented below.
>
> | Shape      | Material      | Parameter        | RP-DiffMPM  | GOP-DiffMPM  | AS-DiffMPM  |
> | ---------- | ------------- | ---------------- | --------------- | ---------------- | ---------------------- |
> | Droplet    | Newtonian     | μ                | 7.81 ± 3.52     | 6.94 ± 4.41      | **5.87 ± 3.37**        |
> |            |               | κ                | 32.33 ± 3.25    | **21.91 ± 8.13**     | 24.50 ± 3.98       |
> | Letter     | Newtonian     | μ                | 8.12 ± 3.49     | 7.22 ± 2.38      | **6.05 ± 1.35**        |
> |            |               | κ                | 31.01 ± 5.40    | 23.12 ± 2.17     | **22.36 ± 6.04**       |
> | Cream      | Non-Newtonian | μ                | 19.42 ± 7.61    | 18.67 ± 2.53     | **14.73 ± 5.12**       |
> |            |               | κ                | 46.50 ± 12.09    | 44.12 ± 3.88     | **38.08 ± 4.65**       |
> |            |               | τY               | 6.92 ± 0.58     | 6.58 ± 0.47      | **4.97 ± 0.34**        |
> |            |               | η                | 54.88 ± 12.72    | 46.41 ± 5.85     | **40.01 ± 7.43**       |
> | Toothpaste | Non-Newtonian | μ                | 20.05 ± 5.45    | 19.18 ± 1.29     | **15.30 ± 1.06**       |
> |            |               | κ                | 44.44 ± 4.33    | 44.99 ± 3.04     | **39.72 ± 3.89**       |
> |            |               | τY               | 7.35 ± 1.61     | 6.97 ± 0.52      | **5.12 ± 0.38**        |
> |            |               | η                | 62.19 ± 19.06    | 47.55 ± 21.95     | **41.17 ± 9.01**       |
> | Torus      | Elastic       | E                | **1.85 ± 0.83**    | 2.87 ± 0.75     | 2.69 ± 0.62        |
> |            |               | ν                | 0.22 ± 0.02     | 0.25 ± 0.02      | **0.19 ± 0.01**        |
> | Bird       | Elastic       | E                | **2.31 ± 0.90**    | 2.94 ± 0.81     | 2.51 ± 0.69        |
> |            |               | ν                | 0.22 ± 0.02     | 0.26 ± 0.02      | **0.21 ± 0.01**        |
> | Playdoh    | Plasticine    | E                | 17.38 ± 1.14    | 15.89 ± 1.03     | **11.62 ± 0.89**       |
> |            |               | ν                | 8.31 ± 3.03     | 9.27 ± 2.02      | **6.22 ± 2.01**        |
> |            |               | τY               | 9.91 ± 1.46     | 8.70 ± 0.62      | **6.74 ± 0.51**        |
> | Cat        | Plasticine    | E                | 18.11 ± 1.20    | 16.32 ± 1.05     | **12.01 ± 0.95**       |
> |            |               | ν                | 8.30 ± 2.71     | 8.28 ± 1.02      | **6.23 ± 1.98**        |
> |            |               | τY               | 10.02 ± 0.84    | 9.14 ± 0.71      | **7.20 ± 0.57**        |
> | Trophy     | Granular      | θ_fric            | 2.19 ± 0.15     | 1.95 ± 0.12      | **1.24 ± 0.09**        |
>
> ## (6) Further experiments
>
> Moreover, please refer to answer to point **Q3** of reviewer **rXMc** for results on planar colliders.
>
> > [Q2]. Per Weakness 2, could the authors present results of system identification on elastic and plastic objects?
>
> We present below the results for elastic and plasticine materials using our AS-DiffMPM method and the baselines. Each method was evaluated on six experiments: two material types and with three colliders. Results are averaged over the colliders. Interestingly, all methods generally perform well on the elastic material. This may be due to the simpler nature of the interaction between the elastic material and the collider—since the material tends to bounce away upon contact, the particle-wise collision handling mechanism of AS-DiffMPM has less impact. Indeed, in these scenarios, no fine-grained interaction (e.g., material flowing around sharp corners) occurs.
>
> | Material       | Parameter     | RP-DiffMPM   | GOP-DiffMPM  | AS-DiffMPM  |
> | -------------- | ------------- | ------------ | ------------ | ----------------- |
> | **Elastic**    | E             | 2.01 ± 0.45 | 2.87 ± 1.18 | **1.91 ± 0.96**   |
> |                | ν             | **0.36 ± 0.43**  | 1.23 ± 0.32  | 0.49 ± 0.12   |
> | **Plasticine** | E             | 28.45 ± 3.12 | 16.88 ± 1.84 | **12.07 ± 2.21**  |
> |                | ν             | 9.29 ± 5.04  | 10.24 ± 3.03  | **8.21 ± 1.02**   |
> |                | τY            | 9.44 ± 2.05  | 8.76 ± 1.88  | **6.23 ± 1.47**   |
>
> > [Q3]. Per Weakness 3, [...]
>
> Please see answer to point **W1** of reviewer **dazL**.
>
> > [Q4].How long does it take (on average) to finish [...]
>
> Please see answer to point **Q4** of reviewer **rXMc**.
>
>
> [1] Xuan Li et al. PAC-NeRF: Physics Augmented Continuum Neural Radiance Fields for Geometry-Agnostic System Identification. In International Conference on Learning Representations (ICLR) 2023.

---

> > ### Comment · Reviewer_CpqV · 2025-08-03
> >
> > Thank you for the response. While I appreciate the authors' effort in supplementing experimental results, I still have the following questions and concerns:
> >
> > 1. In the response [Q1], what are the units for different parameters, e.g., $\mu$ and $\kappa$? Are they on a log scale?
> >
> > 2. My concern on Weakness 3 remains after reading the reply [W1] to **dazL**.
> > Specifically, the current manuscript does not adequately demonstrate the need for supporting non-planar colliders in system identification tasks within a *synthetic* environment. In this case, we have complete control over the simulation assets, and I see no compelling reason to complicate the task unnecessarily when we can simply use planar colliders for system identification.
> > However, I would agree that this ability is important for real-world applications. Often, real-world data is captured from unique, irreproducible events (e.g., structural failure) or in uncontrolled environments where interacting surfaces are inherently complex and non-planar. In these scenarios, we must analyze the event as it happened; re-capturing the phenomenon with a simplified planar collider is frequently impractical or outright impossible due to various constraints.
> > Given this, I think real-world comparisons are essential to validate the utility of the proposed method. The authors claim that it may be challenging to conduct real-world experiments for fluid or granular objects. However, it is possible to consider elastic or plastic objects, as has been demonstrated in previous work [4,35].

---

> > > ### Author Response · Authors · 2025-08-06
> > > **Experiment on real-world data**
> > >
> > > > [C1]. In the response [Q1], what are the units for different parameters, e.g., μ and κ? Are they on a log scale?
> > >
> > > Sorry for the confusion. Yes, they are all on a log scale, except for θ_fric.
> > >
> > > > [C2]. My concern on Weakness 3 remains [...]
> > >
> > > Following the reviewer’s suggestion, we conducted a real-world experiment using a dough sample roughly the size of a football. We modeled the dough as a **non-Newtonian material**.
> > >
> > > The experiment was performed as follows: we released the dough in free fall and let it collide with the **edge of a box**, a scenario designed to create a non-planar contact surface. Upon impact, part of the dough adhered to the **horizontal surface**, while the rest deformed and gradually flowed along the **vertical edge**.
> > >
> > > We captured the event using **five cameras** placed around the scene. The segmentation of the dough in each frame was performed using SAMv2 [1], and the resulting masks were used during optimization.
> > >
> > > The overall system identification pipeline works in three different step:
> > >
> > > ### Step 1: Static Geometry Reconstruction
> > >
> > > We used 2DGS [2] with isotropic Gaussians on the first frame of each view to reconstruct the dough’s initial geometry, followed by internal Gaussians infilling [3] and fine-tuning using the whole set of Gaussians. During fine-tuning, we disabled densification and pruning to preserve the Gaussians' layout. These adjustments resulted in more realistic MPM simulation.
> > >
> > > ### Step 2: Initial Motion Vectors Optimization
> > >
> > > We manually annotated the frame before collision and estimate the **initial velocity** and **gravity vectors** from visual observations during the free-fall phase.
> > >
> > > ### Step 3: Physical Parameter Optimization
> > >
> > > With the velocity and gravity vectors fixed, we optimized the **physical parameters** of the dough from visual observations over the full sequence.
> > >
> > > ---
> > >
> > > ## Results
> > >
> > > The final estimated parameters were:
> > >
> > > ```
> > > Non-Newtonian material with parameters:
> > > μ       = 57033.1
> > > κ       = 139650.4
> > > τY      = 7919.1
> > > η       = 25.1
> > > ```
> > >
> > > Moreover, we evaluated the visual reconstruction quality for three seconds after collision. Below, we report the reconstruction metrics after one, two, and three seconds. Each table presents metrics averaged over 60 frames (i.e., one second at 60 FPS) and 5 views:
> > >
> > > ```
> > > After 1 second:
> > > PSNR  = 33.2 ± 4.1
> > > SSIM  = 0.994 ± 0.004
> > > LPIPS = 0.019 ± 0.001
> > > ```
> > >
> > > ```
> > > After 2 seconds:
> > > PSNR  = 28.9 ± 6.6
> > > SSIM  = 0.988 ± 0.011
> > > LPIPS = 0.024 ± 0.011
> > > ```
> > >
> > > ```
> > > After 3 seconds:
> > > PSNR  = 25.3 ± 7.8
> > > SSIM  = 0.966 ± 0.013
> > > LPIPS = 0.035 ± 0.012
> > > ```
> > >
> > > We observed a degradation in image reconstruction quality during the final frames of the sequence. As the object undergoes significant deformation, the Gaussians deviate substantially from their original positions, introducing visual artifacts. We expect this problem to be mitigated by modeling the Gaussians using time-dependent attributes (e.g., spherical harmonics, rotations). However, we leave this analysis for future work.
> > >
> > > ---
> > >
> > > In conclusion, although NeurIPS rebuttal guidelines prevent from attaching images or videos, we will include such analysis—along with visual comparisons between the reconstructed and ground-truth observations—in the final version of our manuscript.
> > >
> > > We thank the reviewer for their valuable feedback and remain open to further questions or clarifications.
> > >
> > > ---
> > >
> > >
> > > [1] Ravi, Nikhila et al., SAM 2: Segment Anything in Images and Videos, ICLR 2025
> > >
> > > [2] Huang, Binbin et al., 2d gaussian splatting for geometrically accurate radiance fields, ACM SIGGRAPH 2024
> > >
> > > [3] Xie, Tianyi et al., PhysGaussian: Physics-Integrated 3D Gaussians for Generative Dynamics, CVPR 2024

---

> > > > ### Comment · Reviewer_CpqV · 2025-08-08
> > > >
> > > > Thank you for providing results from a real-world scenario, which effectively addresses my primary concern regarding the practicality of the proposed method. Based on the authors' responses, I have no further questions and would like to raise my score to 4. Please ensure that all materials provided during the rebuttal period are included in the final paper.

---

> > > > > ### Author Response · Authors · 2025-08-08
> > > > >
> > > > > We will include the material in the final paper.
> > > > > Thank you for your feedback — we are glad that our response has effectively addressed your concerns.

---

### Official Review · Reviewer_gepN · 2025-06-26

**Clarity:** 2
**Significance:** 3
**Originality:** 3
**Rating:** 4
**Confidence:** 4

**Summary:**

This paper introduces AS-DiffMPM, a differentiable Material Point Method framework that supports system identification of continuum objects colliding with arbitrarily shaped rigid bodies. Unlike prior works that only handle planar boundaries, AS-DiffMPM incorporates a particle-wise collision handling mechanism—based on Compatible Particle-in-Cell (CPIC)—to resolve interactions with complex meshes or 2D Gaussian splat primitives. The method is integrated with both voxel-based NeRF and point-based Gaussian splatting renderers to optimize physical parameters from multi-view videos. Extensive experiments demonstrate its effectiveness on Newtonian, non-Newtonian, and granular materials, using both ground-truth particle trajectories and visual observations for supervision

**Questions:**

See Weakness.

**Ethical Concerns:**

["NO or VERY MINOR ethics concerns only"]

**Final Justification:**

Most of concerns are to some extent addressed.

**Limitations:**

See Weakness.

**Paper Formatting Concerns:**

Appendix can be directly added to the end of the paper.

**Quality:**

3

**Strengths And Weaknesses:**

Strengths

1.Extends differentiable MPM to arbitrary collider geometries via CPIC, enabling accurate, particle-wise velocity corrections during P2G and G2P stages.

2.Provides seamless interfaces to voxel-grid NeRFs and 2D Gaussian Splatting methods, supporting end-to-end gradient flow from photometric losses to physical parameters .

3.Benchmarks across three material types (Newtonian, non-Newtonian, granular) and three colliders (Box, Bunny, Armadillo), reporting both trajectory-based and vision-based identification results, with clear error statistics and comparisons against two baselines (GOP-DiffMPM, RP-DiffMPM) and multiple rendering backends.

4.Authors plan to release code, facilitating reproducibility and further research in differentiable physics and system identification.


Weaknesses
1. MPM is fundamentally a Galerkin-based method—yet the paper does not analyze whether the differentiable formulation preserves energy conservation, admits closed-form solutions, or under what conditions it remains stable.

2. All P2G/G2P and collision computations occur on the background grid; the results are therefore highly dependent on grid resolution, with coarser grids potentially introducing aliasing or inaccuracy.

3. The paper positions CPIC-based collision handling as a more efficient alternative to hard-clamp methods, compensating for approximation errors with speed. However, it never quantifies whether the computational effort saved by avoiding per-particle constraints is fully realized once the overhead of scatter/gather operations and backpropagation through the grid is accounted for. This raises the question of whether any real efficiency advantage survives in practical, high-resolution scenarios.

4. There is no discussion of the method’s numerical properties: e.g., under what time step or grid refinement regimes the optimization converges, whether the gradient signals remain well-conditioned, or if any spurious oscillations arise during backpropagation.

---

> ### Author Rebuttal · Authors · 2025-07-31
>
> We thank the reviewer for the insightful feedbacks on our work. We are happy to know that you recognize the contribution for handling complex geometries within the MPM framework and the results on system identification from both particle trajectories and visual observations.
>
> ***In this rebuttal, we address concerns about the theoretical properties and computational efficiency of AS-DiffMPM. We clarify the distinction between particle-wise and grid-level collision handling, present timing benchmarks across methods, and include a convergence analysis of gradients of physical parameters during optimization.***
>
> We address below every point raised by the reviewer, and we welcome any follow-up question.
>
> > [W1]. MPM is fundamentally a Galerkin-based method—yet the paper does not analyze whether the differentiable formulation preserves energy conservation, admits closed-form solutions, or under what conditions it remains stable.
>
> We thank the reviewer for raising this important point regarding the theoretical properties of our differentiable MPM framework. We have observed empirically stable behavior across all evaluated material types (Newtonian, non-Newtonian, and granular) and collider geometries (Box, Bunny, and Armadillo).
>
> Our implementation builds upon PAC-NeRF [1] and the DiffTaichi framework [2], inheriting their default simulation settings: a timestep of Δt = 0.0002 and a grid spacing of Δx = 0.02. Our code explicitly monitors numerical stability after every simulation step by checking for the **Courant–Friedrichs–Lewy (CFL) condition**: `v_max × Δt ≤ Δx`, where `v_max` is the maximum particle velocity detected at the current timestep. This condition checks if a particle travels more than one grid cell per timestep. If violated, the common approach is to halve the timestep Δt and retry the simulation step.
>
> > [W2]. All P2G/G2P and collision computations occur on the background grid; the results are therefore highly dependent on grid resolution, with coarser grids potentially introducing aliasing or inaccuracy.
>
> We apologize for the confusion. In our paper, collision computations (both **collision check** and **velocity corrections** upon collision) occur entirely on the background grid for the GOP-DiffMPM baseline and in scenarios where the collider has a simple, analytically defined shape (e.g., a plane, as described in lines 122–126 in our manuscript and adopted in previous work [1]). These approaches are highly dependent on grid resolution. In practice, when a collider surface intersects a grid cell, all particles within that cell receive the same velocity update—regardless of their position relative to the surface—which can lead to artifacts such as penetration. These artifacts tend to worsen as the grid resolution decreases.
>
> In contrast, our AS-DiffMPM framework can partially mitigate these issues through **particle-wise** collision handling. Although we still use a background grid to determine which nodes are inside or outside a collider (i.e., **collision check** using the Collision Grid), **velocity corrections** are applied at the **particle** level during the G2P stage, rather than during the GOP stage. This enables control over each particle's interaction with the collider, allowing us finer control such as resolving penetration artifacts (see lines 183–185 in our manuscript). Because these corrections happen on a per-particle basis, AS-DiffMPM is less sensitive to grid resolution compared to grid-level methods.
>
> We hope this response addresses the reviewer's concern. If further clarification is needed, we would be happy to provide it. We will also clarify this point in the revised manuscript.
>
> > [W3]. The paper positions CPIC-based collision handling as a more efficient alternative to hard-clamp methods, compensating for approximation errors with speed. However, it never quantifies whether the computational effort saved by avoiding per-particle constraints is fully realized once the overhead of scatter/gather operations and backpropagation through the grid is accounted for. This raises the question of whether any real efficiency advantage survives in practical, high-resolution scenarios.
>
> We thank the reviewer for raising this important point on computational efficiency, which we acknowledge was not sufficiently analyzed in the original submission.
>
> To address this, we provide below a performance comparison of the collision-handling strategies used in our work: GOP-DiffMPM (**grid-level** collision handling), RP-DiffMPM (collider represented as **rigid particles**), and AS-DiffMPM (our **particle-wise** CPIC-based strategy). Timings are measured per simulation step using a 64x64x64 background grid, 25K particles and during collisions with the Box collider.
>
> ### Timing comparison
> | Method      | Time (ms) |
> | ----------- | --------- |
> | RP-DiffMPM  | 1812    |
> | GOP-DiffMPM | 2192    |
> | AS-DiffMPM  | 2537    |
>
> Moreover, we remind that our AS-DiffMPM extend the MPM by (1) constructing a Collision Grid (Sec. 4.1.1), (2) transfering the Collision Grid to material particles (Sec. 4.1.2), (3) adding compatibility checks during P2G and (4) particle-wise velocity corrections during G2P (Sec. 4.1.3). We report below the time required for these operations.
>
> ### Overhead Breakdown for AS-DiffMPM
> | Component                                                        | Time (ms) |
> | ---------------------------------------------------------------- | --------- |
> | Collision grid construction                                      | 779        |
> | Collision grid to material particles                             | 403        |
> | P2G with compatibility checks                                    | 478        |
> | G2P with particle-wise velocity correction                       | 541        |
>
> Although AS-DiffMPM introduces a modest overhead compared to the baselines, it obtains superior physical accuracy for system identification with complex colliders (see Table 1 in our manuscript), due to its fine-grained collision handling mechanism.
>
> Regarding the mention of “hard-clamp methods”: we clarify that our work does not compare against true hard constraints, but rather grid-level collision correction (as in GOP-DiffMPM), which may lead to simulation artifacts (e.g., penetration) in high-curvature scenarios. Our contribution is not focused on raw speed improvements, but rather on enabling a more accurate collision handling through particle-wise velocity corrections.
>
>
> > [W4]. There is no discussion of the method’s numerical properties: e.g., under what time step or grid refinement regimes the optimization converges, whether the gradient signals remain well-conditioned, or if any spurious oscillations arise during backpropagation.
>
> We acknowledge that our original submission did not include an analysis on optimization convergence, and we agree that such analysis would strengthen the empirical evaluation of AS-DiffMPM.
>
> Below, we provide an analysis of the gradient of physical parameters over 150 steps (a subset of steps is visualized) during system identification using AS-DiffMPM. Specifically, we report results from 3 representative experiments, one per material, colliding against a Box shape.
>
> ### Newtonian material
> | Step  | 1   | 5   | 10  | 20  | 40  | 60  | 90  | 120 | 150 |
> | ----- | --- | --- | --- | --- | --- | --- | --- | --- | --- |
> | **μ** | -0.02 | 0.005 | 0.01 | -0.02 | 0.001 | -0.001 | -0.0001 | -2.9e-05 | -2.4e-06 |
> | **κ** | -0.002 | -0.0008 | 0.0001 | -0.0001 | -0.0002 | -0.0001 | -0.0001 | 4.4e-05 | 5.8e-06 |
>
> ### Non-Newtonian material
> | Step  | 1   | 5   | 10  | 20  | 40  | 60  | 90  | 120 | 150 |
> | ----- | --- | --- | --- | --- | --- | --- | --- | --- | --- |
> | **μ** | -0.004 | -0.004 | -0.004 | 5.1e-05 | -0.0004 | 0.0002 | -3.7e-05 | -1.5e-05 | -1.5e-05 |
> | **κ** | -0.0005 | -0.01 | -0.02 | 0.003 | 0.0004 | 0.002 | 0.0004 | 0.0002 | 0.0002 |
> | **τY** | -0.002 | -0.01 | -0.02 | 0.004 | -0.009 | 0.005 | -0.001 | -0.0003 | -0.0002 |
> | **η** | -0.008 | -0.01 | -0.02 | -0.003 | 0.001 | -0.0002 | 0.0001 | -7.5e-05 | -7.7e-05 |
>
>
> ### Granular material
> | Step  | 1   | 5   | 10  | 20  | 40  | 60  | 90  | 120 | 150 |
> | ----- | --- | --- | --- | --- | --- | --- | --- | --- | --- |
> | **θ_fric** | -0.004 | -0.003 | -0.002 | -0.0006 | 9.7e-05 | 9.1e-05 | 5.8e-05 | 1.6e-05 | -3.3e-06 |
>
>
> The tables above illustrate how the gradients of the physical parameters consistently decrease in magnitude during optimization, suggesting that no problematic oscillations arise to prevent smooth training. While we acknowledge that a more in-depth numerical analysis would be beneficial, we did not observe any stability issues during our empirical evaluations. We therefore leave this study to future work.
>
>
> [1] Xuan Li et al. PAC-NeRF: Physics Augmented Continuum Neural Radiance Fields for Geometry-Agnostic System Identification. In International Conference on Learning Representations (ICLR) 2023.
>
> [2] Yuanming Hu et al. DiffTaichi: Differentiable Programming for Physical Simulation. In International Conference on Learning Representations (ICLR) 2020.

---

> > ### Comment · Reviewer_gepN · 2025-08-02
> > **Rate the score to 4**
> >
> > Thank the author to address most of my concern, I rate my score to 4.

---

> > > ### Author Response · Authors · 2025-08-07
> > >
> > > Thanks for your feedback and we are pleased that our response has successfully addressed your concerns.

---

### Official Review · Reviewer_rXMc · 2025-06-28

**Clarity:** 3
**Significance:** 3
**Originality:** 3
**Rating:** 4
**Confidence:** 3

**Summary:**

This paper introduces AS-DiffMPM, a differentiable Material Point Method (MPM) framework that supports system identification in scenes involving interactions between deformable objects and arbitrarily shaped rigid body colliders. The authors extend differentiable MPM techniques with a particle-wise collision handling mechanism inspired by CPIC, enabling gradient-based physical parameter estimation in the presence of non-planar, complex rigid bodies. AS-DiffMPM is evaluated in combination with various rendering pipelines (voxel-based NeRF, 2DGS, MDyn-3DGS), and both qualitative and quantitative results demonstrate its effectiveness across multiple material types and collider

**Questions:**

1. In Table 1 (System Identification from Particle Trajectories), AS-DiffMPM appears to outperform the baselines predominantly in the case of Non-Newtonian materials, whereas for Newtonian and Granular materials, the baseline methods achieve lower errors in many settings. Could the authors elaborate on the underlying reasons for this discrepancy? What characteristics of Non-Newtonian materials make AS-DiffMPM particularly effective?
2. In Table 3 (Geometric Error Evaluation), AS-DiffMPM only outperforms baselines in the case of Granular materials, which does not fully align with the trends observed in Table 1 for System Identification. Could the authors provide a deeper analysis of under what conditions (material type, collider complexity, or metric) AS-DiffMPM provides superior performance compared to GOP-DiffMPM or RP-DiffMPM?
3. The proposed method is titled Any-Shape Differentiable Material Point Method, yet no experimental results are reported for interactions with planar colliders—scenarios where prior works (e.g., PAC-NeRF, GIC) are typically evaluated. Could the authors clarify whether AS-DiffMPM maintains competitive performance in this condition?
4. The method is evaluated in conjunction with several rendering pipelines (DVGO, 2DGS, and MDyn-3DGS), which differ substantially in complexity and representation. Could the authors quantify the differences in training time and rendering latency among these models in the context of the system identification task? A breakdown of computational efficiency would be helpful for practitioners deciding between rendering backends.

**Ethical Concerns:**

["NO or VERY MINOR ethics concerns only"]

**Final Justification:**

The authors clarify the effectiveness of the proposed method for non-Newtonian materials, the trend difference between Table 1 and Table 3, and the training times, addressing my concerns.

I keep my rate "borderline accept" after the rebuttal and discussion.

**Limitations:**

Yes. The paper explicitly addresses its limitations, most notably the assumption of static rigid bodies and the absence of real-world evaluations. The authors also mention that their approach could be extended to moving colliders but defer such experimentation to future work. This upfront discussion is appreciated.

**Quality:**

3

**Strengths And Weaknesses:**

Strengths
1. [Exploration of non-planar colliders] The paper addresses a previously underexplored problem—system identification with non-planar rigid body colliders—by extending differentiable MPM to handle complex geometry interactions. This is a novel and necessary advancement over prior works like PAC-NeRF and GIC. By supporting arbitrarily shaped colliders, AS-DiffMPM enables more realistic physics-based analysis and simulation. This has clear value for downstream applications in robotics, graphics, and 3D perception.
2. [Good compatibility] AS-DiffMPM is designed to be compatible with multiple rendering pipelines, increasing its adaptability and potential for broad adoption.
3. [Technical novelty of Integrating CPIC] The proposed method integrate CPIP to project the rigid body onto a grid-based distance field so that it can handle collisions in a particle-wise manner and apply velocity corrections on particles in local neighborhood.
4. [Extensive experimental evaluation] The authors conduct extensive evaluations on both particle-trajectory-based and vision-based system identification tasks across diverse material types and collider geometries, showing superior or competitive performance.


Weaknesses
1. [No comparison with existing method on planar colliders] Although the paper emphasizes AS-DiffMPM’s capability to handle interactions with arbitrarily shaped rigid bodies, it does not report results on planar colliders—a standard experimental setting used in prior work such as PAC-NeRF. This omission makes it difficult to assess whether the proposed method remains competitive in the regimes where earlier methods perform well, and prevents a direct comparison under identical conditions.
2. [Lack of qualitative comparison with other methods] The qualitative visualizations provided do not include side-by-side comparisons between AS-DiffMPM, baseline methods (e.g., GOP-DiffMPM), and ground truth. Such comparisons are critical for visually assessing the fidelity of simulated trajectories and would significantly strengthen the empirical claims of the paper.

---

> ### Author Rebuttal · Authors · 2025-07-31
>
> We thank the reviewer for the detailed reading of our paper and constructing suggestions. Specifically, we appreciated the recognition of the impact of the proposed method for handling complex colliders and the extensive evaluation.
>
> ***Briefly, in this rebuttal, we provide additional experiments in the planar setting, clarify the distinct challenges posed by non-Newtonian versus Newtonian and granular materials, and present a computational efficiency analysis of different rendering backends.***
>
> We hope our responses below adequately address your concerns and we are open to further clarification.
>
> > [W1]. [No comparison with existing method on planar colliders] [...]
>
> Please see Q3.
>
> > [W2]. [Lack of qualitative comparison with other methods] [...]
>
> We thank the reviewer for the suggestion and agree that qualitative side-by-side comparisons would further support our claims. As per NeurIPS rebuttal guidelines, we cannot include additional media at this stage, but we plan to add such visualizations to the supplementary material.
>
> > [Q1]. In Table 1 (System Identification from Particle Trajectories), AS-DiffMPM appears to [...]. What characteristics of Non-Newtonian materials make AS-DiffMPM particularly effective?
>
> We agree with the reviewer that this analysis requires further clarification. Firstly, we remind that GOP-DiffMPM operates at the grid level during the Grid Operations (G-OP) stage. It applies a single, **averaged** velocity correction to all particles within a grid cell that is detected to be inside a collider. This is a coarse approximation that may overlook fine-grained interactions between the material and the complex collider.
>
> In contrast, our AS-DiffMPM performs **particle-wise** collision handling during the Particle-to-Grid (P2G) and Grid-to-Particle (G2P) stages. It identifies individual particles near a boundary and applies **particle-wise** velocity corrections based on their specific relationship with the collider surface. These differing collision strategies have practical implications for system identification, particularly depending on the material behavior.
>
> For **fluid-like materials** (e.g., Newtonian fluids or granular materials), where particles tend to slide along the surface or flow away upon impact, the smoothing effect of GOP-DiffMPM’s grid-based velocity correction is often sufficient. Although it may introduce minor inaccuracies at sharp corners (e.g., slight particle penetration), the approximation can contribute to smoother gradients and potentially help stabilize the optimization process. This may partially explain why GOP-DiffMPM performs competitively—or even slightly better—in some Newtonian and granular settings. However, for **non-Newtonian materials** with higher yield stress (τY) [1], which exhibit **solid-like behavior**, this averaging becomes problematic. In such cases, particles are more prone to adhere to surfaces or accumulate near sharp corners. GOP-DiffMPM’s **grid-level smoothing** tends to blur these interactions, causing the simulated material to partially "bleed" into the collider geometry. This results in reduced accuracy when estimating physical parameters. In contrast, AS-DiffMPM’s **particle-wise** correction accurately resolves per-particle contact behavior, particularly around complex or high-curvature boundaries, resulting in better system identification results.
>
> We acknowledge that such nuances were not mentioned in the original manuscript and we will revise it accordingly.
>
> > [Q2]. In Table 3 (Geometric Error Evaluation), AS-DiffMPM only outperforms baselines in the case of Granular materials, which does not fully align with the trends observed in Table 1 for System Identification [...]
>
> We understand the reviewer's concern and acknowledge that this analysis was not throughly discussed.
>
> There is a conceptual distinction between the metrics used in Table 3 (geometric error) and those in Table 1 (parameter estimation error), which explains why they can lead to seemingly different conclusions.
>
> The Chamfer Distance (CD) and Earth Mover’s Distance (EMD) in Table 3 measure the geometric similarity between predicted and ground-truth particle configurations. These metrics compute spatial correspondences between particles in one point cloud and particles in the other point cloud, and quantify how closely the predicted point cloud matches the reference one. However, a low geometric error does not necessarily imply that the correct physical parameters were identified. Two simulations may produce geometrically similar shapes despite being governed by different underlying physics. Specifically, there can be similar shapes but with different particle trajectories. Therefore, upon convergence of the physical parameters estimation, the model might have different particle trajectories but similar shape with respect to the reference, thus better geometric similarity but worse parameters estimation.
>
> We acknowledge that the results and analysis in Table 3 are not properly discussed and we will revise this in our manuscript.
>
> > [Q3]. The proposed method is titled Any-Shape Differentiable Material Point Method [...]
>
> We thank the reviewer for this observation and agree that evaluating AS-DiffMPM under planar collider conditions provides valuable insight into its performance in standard benchmarks. To this end, we conducted additional experiments comparing our AS-DiffMPM with both the baselines used in our paper and the **Standard MPM (S-MPM)** collision strategy adopted by prior works [2, 3, 4]. In S-MPM, collisions are resolved at the **Grid Operations** stage (Sec. 3.1): for a planar surface defined by point **x_b** and normal **n_b**, the nodal velocity **v_g** is set to zero for all grid nodes **x_g** such that **(x_g - x_b) · n_b ≤ 0**. In contrast, AS-DiffMPM represents the same planar collider as a mesh surface positioned at **x_b**, and applies **particle-wise collision handling** during both P2G and G2P steps. We performed three system identification experiments using particle trajectory supervision under planar collider conditions. The results are summarized in the following table, reporting the absolute error in physical parameter estimation.
>
> | Material       | Parameter | S-MPM | RP-DiffMPM | GOP-DiffMPM | AS-DiffMPM (Ours) |
> |----------------|-----------|--------|--------------|-------------|-------------|
> | Newtonian      | µ         | **7.51**    | 8.03          | 7.63         | 7.98         |
> |                | κ         | **31.98**    | 35.21          | 32.45         | 33.21         |
> | Non-Newtonian  | µ         | 24.09    | 24.57          | 24.91         | **24.03**         |
> |                | κ         | **19.01**   | 21.12          | 20.19         |  19.03         |
> |                | τY       | 4.91    | 5.12          | 4.78         | **4.54**         |
> |                | η         | **30.53**    | 32.1          | 31.05         | 31.21         |
> | Granular       | θ_fric    | 0.15    | 0.56          |    0.22      |    **0.12**      |
>
> The S-MPM achieves slightly lower errors for some parameters, likely due to its use of analytically defined boundary conditions, which may offer improved numerical stability in simple geometries. However, a more systematic analysis would be required.
>
> > [Q4]. The method is evaluated in conjunction with several rendering pipelines (DVGO, 2DGS, and MDyn-3DGS) [...]
>
> We report the average time for the following components, measured on an NVIDIA RTX 3080 GPU: (1) full training, (2) a forward pass (i.e., particle advection using MPM followed by rendering), and (3) a backward pass. Full training consists of 150 optimization steps. At each step, we perform a forward and backward pass over all 16 frames in the video, followed by an update of the physical parameters (gradient optimization step).
>
> | Rendering Backend | Full Training (h) | Forward Pass (s) | Backward Pass (s) |
> |-------------------|-------------------|------------------------------|-------------------------------|
> | DVGO              | **2.50**              | **12**                           | **48**                            |
> | 2DGS              | 3.21              | 14                           | 63                            |
> | MDyn-3DGS         | 8.88              | 33                           | 180                           |
>
>
> While MDyn-3DGS has generally demonstrated higher performance than the other methods (see Tab. 2 in our manuscript), it comes at a significantly higher computational cost. This is primarily due to its particle infilling strategy, which increases the number of particles in the MPM simulation, and the use of a loss function that combines both image reconstruction error (between rendered and ground truth masks) and geometric error on the particles. Additionally, the reported timing for MDyn-3DGS does not include the preliminary dynamic scene reconstruction stage required before physical parameter optimization. This preprocessing step is briefly discussed in lines 297–299 of our manuscript; for further details, we kindly refer to [4].
>
>
> [1] Yonghao Yue et al. Continuum foam: A material point method for shear-dependent flows. In ACM Transactions on Graphics (TOG) 2015.
>
> [2] Xuan Li et al. PAC-NeRF: Physics Augmented Continuum Neural Radiance Fields for Geometry-Agnostic System Identification. In International Conference on Learning Representations (ICLR) 2023.
>
> [3] Takuhiro Kaneko. Improving Physics-Augmented Continuum Neural Radiance Field-Based Geometry-Agnostic System Identification with Lagrangian Particle Optimization. In Computer Vision and Pattern Recognition (CVPR) 2024.
>
> [4] Junhao Cai et al. GIC: Gaussian-Informed Continuum for Physical Property Identification and Simulation. In Advances in neural information processing systems (NeurIPS) 2024

---

> > ### Comment · Reviewer_rXMc · 2025-08-06
> >
> > Thank the authors for clarifying the effectiveness of the proposed method for non-Newtonian materials, the trend difference between Table 1 and Table 3, and the training times.
> >
> > Regarding the comparison with previous methods under planar collider conditions, it sounds reasonable that S-MPM achieves slightly lower errors primarily due to the analytically defined boundary conditions. It would be great if the authors provide more analysis in their camera-ready version if the paper is accepted.

---

> > > ### Author Response · Authors · 2025-08-07
> > > **Analysis of analytically-defined boundary conditions**
> > >
> > > Thanks for your feedback and we are pleased that our response has successfully addressed your concerns.
> > >
> > > We will include such analysis in the final version of our manuscript.

---

### Official Review · Reviewer_dazL · 2025-06-29

**Clarity:** 4
**Significance:** 3
**Originality:** 3
**Rating:** 4
**Confidence:** 4

**Summary:**

The authors propose a method to recover physical properties from video for fluid or granular media. The approach combines existing MP methods for physics simulation with differrentiable rendering. Different to prior work, arbitrary colliders (beyond planes) are supported. The contributions are on how to relate these colliders with the grid&point representations in MPM. Notably the reconstruction is done jointly with 3D reconstruction of the shape through differentiable rendering, without an intermediate step to track visual features.

**Questions:**

- how important are tne #of sequences used, when does it degrade?
- are there other limitations, have other material types been tested? Would certain shapes be problematic?
- why is the collider shape imported and not reconstructed from multi-view?

**Ethical Concerns:**

["NO or VERY MINOR ethics concerns only"]

**Final Justification:**

Authors clarified my concerns, I stick to recommending acceptance.

**Limitations:**

- See questions above, I find the current limitation section a little shallow as it adresses only once conceptual issue but leavs out other.

**Paper Formatting Concerns:**

Generally well written but perhaps at times taking a few to many steps back (intro and perhaps introduction of MPM), leaving little space for other details (see self-contained comment above).

**Quality:**

4

**Strengths And Weaknesses:**

Strengths:
- the implementation and experiments are extensive, including both Voxel-based NeRF and Gaussian Splatting as point-based methods.
- experiments are on established benchmarks and i comparison to existing methods

- the addition of arbitrary colliders appears as rather a technical detail yet there are a few considerations to be made and the evaluation is decent, so there is still value for the community.

Weaknesses:
- experiments are only on synthetic sequences, would you think it generalizes to real scenes? Should be included in limitation section.
- It would be better to up-front explain that the focus is on Newtonian fluids, non-Newtonian fluids, and granular media. From the intro I was more expecting deformable solid objects to be tackled, as it was very general.
- in the eperiment section authors refer largely to prior work. To be self contained, it would be beneficial to know how close the initial parameter guesses are to the GT etc.

---

> ### Author Rebuttal · Authors · 2025-07-31
>
> We thank the reviewer for their thoughtful and constructive feedback. We appreciate the recognition of our paper’s quality, clarity, technical soundness and experimental evaluation. Below, we address each of the reviewer’s points in detail.
>
> ***In summary, this rebuttal clarifies the limitations and open challenges that need to be addresses to extend our approach to real world scenarios, and provides additional experiments on novel material types and object shapes.***
>
> Please let us know if there is anything we can clarify further.
>
> > [W1]. experiments are only on synthetic sequences, would you think it generalizes to real scenes? Should be included in limitation section.
>
> We agree that the limitations section should discuss this point. Specifically, a key difficulty lies in the nature of the materials we target — Newtonian, non-Newtonian, and granular — which lack a **fixed rest shape**. This differs from previous work on deformable solids [1], where shape reconstruction and parameter estimation can be decoupled—for example, by first reconstructing the **object's rest shape** and then estimating physical parameters based on **observed deformations**. Instead, when dealing with fluid or granular materials in real-world settings, these tasks must be addressed simultaneously, as the object's shape evolves continuously during interaction. For example, estimating the viscosity of a fluid being poured requires reasoning jointly over its **shape** and **dynamics**.
>
> Our proposed AS-DiffMPM supports such scenarios (e.g., pouring into complex containers; see supplementary video at `supp/SecC2/glass_newtonian.mp4`), and we believe it represents a step forward toward system identification in uncostrained environments (i.e., involving complex colliders). However, to fully achieve system identification of fluid or granular materials in the real world, **joint reconstruction and identification** from real world video is required—an open challenge that lies beyond the scope of this work. We acknowledge that this limitation was not stated in the original manuscript and we will update the limitations section to reflect this point.
>
> > [W2]. It would be better to up-front explain that the focus is on Newtonian fluids, non-Newtonian fluids, and granular media. From the intro I was more expecting deformable solid objects to be tackled, as it was very general.
>
> We thank the reviewer for pointing this out. We acknowledge that the introduction frames the work around the general challenge of system identification, without clearly stating our specific focus on fluid and granular materials. We will revise the introduction to explicitly clarify this scope.
>
> > [W3]. in the eperiment section authors refer largely to prior work. To be self contained, it would be beneficial to know how close the initial parameter guesses are to the GT etc.
>
> For a fair analysis, we generated data and ground truth values using the same procedure as in previous work [2]. Therefore, the conditions (i.e., cameras, initial guess, ground truth values) are the same as in [2]. For completeness, for each material, **we show below the initial guess and the different ground truth physical parameters for each sequence**. We remind that there are 25 different sequences and 3 colliders, totaling 75 sequences for our experiments. We will revise the experiments section to clarify this point.
>
> ### Newtonian (initial guess: µ=10, κ=10000)
> | Parameter | 1        | 2         | 3         | 4         | 5         | 6        | 7         | 8        | 9        | 10        |
> | --------- | -------- | --------- | --------- | --------- | --------- | -------- | --------- | -------- | -------- | --------- |
> | µ         | 19.46    | 436.62    | 155.83    | 121.76    | 49.09     | 38.44    | 64.16     | 228.71   | 552.98   | 106.93    |
> | κ         | 56075.55 | 152696.25 | 193525.59 | 257356.05 | 518012.47 | 13772.52 | 358237.13 | 11041.06 | 16789.77 | 112569.73 |
>
> ### Non-Newtonian (initial guess: µ=100, κ=100000, τY=10, η=1)
> | Parameter | 1         | 2         | 3         | 4         | 5         | 6        | 7         | 8         | 9         | 10        |
> | --------- | --------- | --------- | --------- | --------- | --------- | -------- | --------- | --------- | --------- | --------- |
> | µ         | 13209.25  | 65351.08  | 43757.04  | 36027.61  | 19593.71  | 20522.72 | 51549.45  | 121865.90 | 241579.97 | 33764.59  |
> | κ         | 201566.59 | 171054.03 | 249639.94 | 134751.55 | 121836.33 | 14494.30 | 370317.66 | 32859.59  | 30324.98  | 122896.10 |
> | τY        | 1151.42   | 7491.70   | 3964.94   | 5061.12   | 1462.78   | 4153.38  | 3203.67   | 1192.76   | 1251.29   | 4689.16   |
> | η         | 6.68      | 26.69     | 23.27     | 22.31     | 38.83     | 27.24    | 20.43     | 10.27     | 10.62     | 22.89     |
>
> ### Granular (initial guess: θ_fric=10)
> | Parameter | 1       | 2       | 3       | 4       | 5       |
> | --------- | ------- | ------- | ------- | ------- | ------- |
> | θ_fric    | 30.6577 | 32.3751 | 26.8816 | 29.3458 | 42.2861 |
>
> > [Q1]. how important are tne #of sequences used, when does it degrade?
>
> We would kindly ask the reviewer to clarify what is meant by “number of sequences used”. As described in Sec. 5 "Datasets" and "Physical Parameters", our dataset follows the protocol of [2]: we use 10 sequences each for Newtonian and Non-Newtonian materials, and 5 for Granular, repeated across 3 colliders, totaling 75 sequences. Each sequence is a multi-view video recorded from 11 cameras, and the optimization is performed independently per sequence.
> If the question refers to how performance scales with fewer camera views, we agree this is a relevant research direction. However, this analysis is beyond the scope of our current work. We kindly refer the reviewer to [3] for such analysis.
>
> > [Q2]. are there other limitations, have other material types been tested? Would certain shapes be problematic?
>
> Please refer to our answers to reviewer **CpqV** for additional experiments in this direction. Specifically, we address **material shape variation** and further **material types** in answers to points **Q1** and **Q2** of reviewer **CpqV**.
>
> > [Q3]. why is the collider shape imported and not reconstructed from multi-view?
>
> We thank the reviewer for raising this concern. Sec. 5.1 and 5.2 aims to quantitatively assess the system capabilities in recovering physical parameters. For a fair analysis, all other sources are set at their ideal condition. Thus, we import the collider shape instead of reconstructing it from multiple views since the reconstructed collider might be imprecise, this affects the particle trajectories and thus the estimation of physical parameters. We invite the reviewer to check Sec. B in the supplementary PDF, as we believe it addressess the concern raised here (i.e., it analyzes *the performance drop when using the reconstructed collider instead of the imported one*). We are open to further clarification if needed.
>
>
> [1] Licheng Zhong, Hong-Xing Yu, Jiajun Wu and Yunzhu Li. Reconstruction and Simulation of Elastic Objects with Spring-Mass 3D Gaussians. In European Conference on Computer Vision (ECCV) 2024.
>
> [2] Xuan Li, Yi-Ling Qiao, Peter Yichen Chen, Krishna Murthy Jatavallabhula, Ming Lin, Chenfanfu Jiang and Chuang Gan. PAC-NeRF: Physics Augmented Continuum Neural Radiance Fields for Geometry-Agnostic System Identification. In International Conference on Learning Representations (ICLR) 2023.
>
> [3] Takuhiro Kaneko. Improving Physics-Augmented Continuum Neural Radiance Field-Based Geometry-Agnostic System Identification with Lagrangian Particle Optimization. In Computer Vision and Pattern Recognition (CVPR) 2024.

---

> > ### Comment · Reviewer_dazL · 2025-08-05
> >
> > Thanks for the detailed answers.
> >
> > Yes, with # sequences I wondered about how many observations are used for reconstruction. It was not clear to me whether a single simulation was used or several simulations with the same parameters but different initial conditions. Hence, the provided answer partially answers my question. If only a single simulation run is used, it would be important to know how many relevant frames there are in each sequence and what the effect of the sequence length is. The number of views would also be interesting but not so important for this work focussing on the physics.
> >
> > All other points are clarified well.

---

> > > ### Author Response · Authors · 2025-08-07
> > > **Details on simulation sequences**
> > >
> > > Yes, a single simulation run is used.
> > >
> > > Every simulation is 16-frames long and it is made of 11 cameras around the scene. For every simulation, the object hits the collider at frame 6. Thus, frames from 6 to 16 are the most relevant ones. We agree with the reviewer that identifying the most informative frames within the 6 to 16 interval is of interest. However, a detailed analysis would be required, considering also that the relevance of each frame may depend on the material type. We therefore leave this investigation for future work.
> > >
> > > While we have not experimented with longer sequences, we expect that increasing the sequence length could lead to vanishing gradient issues during optimization. However, this limitation could be addressed through truncated backpropagation strategies—for instance, by first optimizing over frames 0 to 15, and then over 15 to 30.
> > >
> > > ---
> > >
> > > Finally, as a follow-up of [W1], we kindly invite the reviewer to read our comment "experiment on real-world data" to reviewer CpqV.
> > >
> > > We thank the reviewer and we are open to further questions or clarifications

---

### Note · Authors · 2025-08-12

In this work, we tackle system identification for a variety of material types interacting with arbitrarily shaped colliders.

Following the rebuttal phase, we have addressed all reviewer concerns and questions, and reviewers were happy to raise their score.

Notably, we conducted a real-world experiment, which further strengthens our contribution and demonstrates the practical importance of modeling complex colliders for system identification in real-world settings. We also carried out additional experiments, such as diverse collider conditions (e.g., shape, size, position) and further material types (elastic and plastic) and shapes (e.g., bird, toothpaste).
Finally, we provided a runtime analysis and additional clarifications, such as the distinct challenges posed by different material types.

We sincerely thank the reviewers for their valuable feedback, which has helped to further strengthen our work.

---

### Decision · Program_Chairs · 2025-09-17

**Decision:**

Accept (poster)

**Comment:**

The paper received 4 reviews. The authors-reviewers discussion clarified most points raised in the initial reviews. A consensus that the paper brings a valuable contribution with sufficient validation was reached; the paper can be revised to meet the requirements for publication based on the authors-reviewers discussion. The scores were revised accordingly. The paper could be accepted as poster.